# GAGA: Deciphering Age-path of Generalized Self-paced Regularizer

**Xingyu Qu**[1*]  **Diyang Li**[2*]  **Xiaohan Zhao**[2]  **Bin Gu**[1,2†]

[1] Mohamed bin Zayed University of Artificial Intelligence
[2] Nanjing University of Information Science & Technology
{Xingyu.Qu,bin.gu}@mbzuai.ac.ae,
Diyounglee@gmail.com, xiaohan.zhao42@foxmail.com

## Abstract

Nowadays self-paced learning (SPL) is an important machine learning paradigm that mimics the cognitive process of humans and animals. The SPL regime involves a self-paced regularizer and a gradually increasing age parameter, which plays a key role in SPL but where to optimally terminate this process is still non-trivial to determine. A natural idea is to compute the solution path w.r.t. age parameter (*i.e.*, age-path). However, current age-path algorithms are either limited to the simplest regularizer, or lack solid theoretical understanding as well as computational efficiency. To address this challenge, we propose a novel Generalized Age-path Algorithm (GAGA) for SPL with various self-paced regularizers based on ordinary differential equations (ODEs) and sets control, which can learn the entire solution spectrum w.r.t. a range of age parameters. To the best of our knowledge, GAGA is the first *exact* path-following algorithm tackling the age-path for *general* self-paced regularizer. Finally the algorithmic steps of classic SVM and Lasso are described in detail. We demonstrate the performance of GAGA on real-world datasets, and find considerable speedup between our algorithm and competing baselines.

## 1 Introduction

**The SPL.**    Self-paced learning (SPL) [1] is a classical learning paradigm and has attracted increasing attention in the communities of machine learning [2, 3, 4, 5, 6], data mining [7, 8] and computer vision [9, 10, 11]. The philosophy under this paradigm is simulating the strategy that how human-beings learn new knowledge. In other words, SPL starts learning from easy tasks and gradually levels up the difficulty while training samples are fed to the model sequentially. At its core, SPL can be viewed as an automatic variant of curriculum learning (CL) [12, 13], which uses prior knowledge to discriminate between simple instances and hard ones along the training process. Different from CL, the SPL assigns a real-valued "easiness weight" to each sample implicitly by adding a self-paced regularizer (SP-regularizer briefly) to the primal learning problem and optimizes the original model parameters as well as these weights. Considering this setting, the SPL is reported to alleviate the problem of getting stuck in bad local minima and provides better generalization as well as robustness for the models, especially in hard condition of heavy noises or a high outlier rate [1, 14].

There are two critical aspects in SPL, namely the SP-regularizer and a gradually increasing age parameter. Different SP-regularizers can be designed for different kinds of training tasks. At the primary stage of SPL, only the hard SP-regularizer is utilized and leads to a binary variable for weighting samples [1]. Going with the advancing of the diverse SP-regularizers [15], SPL

---

*These authors contributed equally.
†Corresponding Author

equipped with different types of SP-regularizers has been successfully applied to various applications [16, 3, 17]. As for the age parameter (*a.k.a.* pace parameter), the users are expected to increase its value continually under the SPL paradigm, given that the age parameter represents the maturity of current model. A lot of empirical practices have turned out that seeking out an appropriate age parameter is crucial to the SPL procedure [18]. The SPL tends to obtain a worse performance in the presence of noisy samples/outliers when the age parameter gets larger, or conversely, an insufficient age parameter makes the gained model immature (*i.e.* underfitting. See Figure 1).

**On the age-path of SPL.** Although the SPL is a classical and widespread learning paradigm, when to stop the increasing process of age parameter in implementation is subject to surprisingly few theoretical studies. In the majority of practices [19], the choice of the optimal model age has, for the time being, remained restricted to be made by experience or by using the trial-and-error approach, which is to adopt the alternate convex search (ACS) [20] multiple times at a predefined sequence of age parameters. This operation is time-consuming and could miss some significant events along the way of age parameter. In addition, the SPL regime is a successive training process, which makes existing hyperparameter tuning algorithms like parallelizing sequential search [21] and bilevel optimization [22] difficult to apply. Instead of training multiple subproblems at different age parameters, a natural idea is to calculate the *solution path* about age parameter, namely age-path (e.g., see Figure 2). A solution path is a set of curves that demonstrate how the optimal solution of a given optimization problem changes w.r.t. a hyperparameter. Several papers like [23, 24] laid the foundation of solution path algorithm in machine learning by demonstrating the rationale of path tracking, which is mainly built on the Karush-Khun-Tucker (KKT) theorem [25]. Existing solution path algorithms involve generalized Lasso [26], semi-supervised support vector classification [27], general parametric quadratic programming [28], etc. However, none of the existing methods is available to SPL regime because they are limited to uni-convex optimization while the SPL objective is a *biconvex* formulation. Assume we've got such an age-path, we can observe the whole self-paced evolution process clearly and recover useful intrinsic patterns from it.

**State of the art.** Yet, a rapidly growing literature [29, 19, 30, 31] is devoted to developing better algorithms for solving the SPL optimization with ideas similar to age-path. However, despite countless theoretical and empirical efforts, the understanding of age-path remains rather deficient. Based on techniques from incremental learning [32], [31] derived an exact age-path algorithm for mere *hard SP-regularizer*, where the path remains piecewise constant. [29, 30] proposed a multi-objective self-paced learning (MOSPL) method to approximate the age-path by evolutionary algorithm, which is not theoretically stable. Unlike previous studies, the difficulty of revealing the exact generalized age-path lies in the continuance of imposed weight and the alternate optimization procedure used to solve the minimization function. From this point of view, the technical difficulty inherent in the study of age-path with general SP-regularizer is intrinsically more challenging.

**Proposed Method.** In order to tackle this issue, we establish a novel Generalized Age-path Algorithm (GAGA) for various self-paced regularizers, which prevents a straightforward calculation of every age parameter. Our analysis is based on the theorem of partial optimum while previous theoretical results are focused on the implicit SPL objective. In particular, we enhance the original objective to a single-variable analysis problem, and use different sets to partition samples and functions by their confidence level and differentiability. Afterward, we conduct our main theorem results based on the technique of ordinary differential equations (ODEs). In the process, the solution path hits, exits, and slides along the various constraint boundaries. The path itself is piecewise smooth with kinks at the times of boundary hitting and escaping. Moreover, from this perspective we are able to explain some shortcomings of conventional SPL practices and point out how we can improve them. We believe that the proposed method may be of independent interest beyond the particular problem studied here and might be adapted to similar biconvex schemes.

**Contributions.** Therefore, the main contributions brought by this work are listed as follows.

- We firstly connect SPL paradigm to the concept of partial optimum and emphasize its importance here that has been ignored before, which gives a novel viewpoint to the robustness of SPL. Theoretical studies are conducted to reveal that our result does exist some equivalence with previous literature, which makes our study more stable.

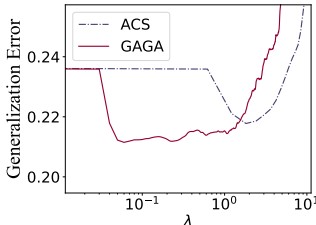

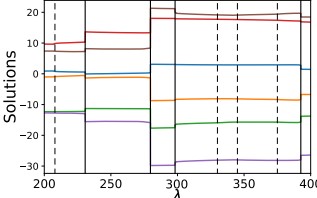

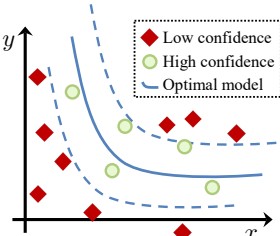

Figure 1: Learning curve against age $\lambda$. The curve is recorded when running linear regression on music dataset.

Figure 2: An age-path visualisation. Different vertical lines represent different types of critical points. The figure is plotted on random 60% features from diabetes dataset using Lasso with $\alpha = 0.01$.

Figure 3: An *example* of set partition in 2-D space. Sample points of same colors belong to one set. The two dashed lines represent partition boundaries (smooth surfaces), which satisfies $l_i = \lambda$.

- A framework of computing the *exact* age-path for *generalized* SP-regularizer is derived using the technique of ODEs, which allows for the time-consuming ACS to be avoided. Concrete algorithmic steps of classic SVM [33] and Lasso [34] are given for implementation.

- Simulations on real and synthetic data are provided to validate our theoretical findings and justify their impact on the designing future SPL algorithms of practical interest.

**Notations.** We write matrices in uppercase (*e.g.*, $X$) and vectors in lowercase with bold font (*e.g.*, $\boldsymbol{x}$). Given the index set $\mathcal{E}$ (or $\mathcal{D}$), $X_{\mathcal{E}}$ (or $X_{\mathcal{E}\mathcal{D}}$) denotes the submatrix that taking rows with indexes in $\mathcal{E}$ (or rows/columns with indexes in $\mathcal{E}/\mathcal{D}$, respectively). Similarly notations lie on $\boldsymbol{v}_{\mathcal{E}}$ for vector $\boldsymbol{v}$, $\boldsymbol{\ell}_{\mathcal{E}}(x)$ for vector functions $\boldsymbol{\ell}(x)$. For a set of scalar functions $\{\ell_i(x)\}_{i=1}^n$, we denote the vector function $\boldsymbol{\ell}(x)$ where $\boldsymbol{\ell}(x) = (\ell_i(x))_{i=1}^n$ without statement and vice versa. Moreover, we defer the full proofs as well as the algorithmic steps on applications to the Appendix.

## 2 Preliminaries

### 2.1 Self-paced Learning

Suppose we have a dataset containing the label vector $\boldsymbol{y} \in \mathbb{R}^n$ and $X \in \mathbb{R}^{n \times d}$, where $n$ samples with $d$ features are included. The $i$-th row $X_i$ represents the $i$-th data sample $x_i$ (*i.e.*, the $i$-th observation). In this paper, the following unconstrained learning problem is considered

$$\min_{\boldsymbol{w} \in \mathbb{R}^d} \sum_{i=1}^{n} \ell\left(x_i, y_i; \boldsymbol{w}\right) + \sum_{j=1}^{m} \alpha_j \mathcal{R}_j(\boldsymbol{w}), \tag{1}$$

where $\mathcal{R}_j(\cdot)$ is the regularization item with a positive trade-off parameter $\alpha_j$, and $\ell_i(\boldsymbol{w})$[3] denotes loss function w.r.t. $\boldsymbol{w}$.

**Definition 1** ($\mathbb{PC}^r$ Function). *Let $f : U \to \mathbb{R}$ be a continuous function on the open set $U \in \mathbb{R}^n$. If $\{f_i\}_{i \in I_f}$ is a set of $\mathbb{C}^r$ (i.e., r-times continuously differentiable) functions such that $f(\boldsymbol{x}) \in \{f_i(\boldsymbol{x})\}_{i \in I_f}$ holds for every $\boldsymbol{x} \in U$, then $f$ is an r-times piecewise continuously differentiable function, namely $\mathbb{PC}^r$ function. The $\{f_i\}_{i \in I}$ is a set of selection functions of $f$.*

**Assumption 1.** *We assume that $\ell_i(\boldsymbol{w})$ and $\mathcal{R}_j(\boldsymbol{w})$ are convex $\mathbb{PC}^r$ functions each with a set of selection functions $\bigcup_{k \in I_{\ell_i}} \{D_{\ell_i}^k\}$ and $\bigcup_{k \in I_{\mathcal{R}_j}} \{D_{\mathcal{R}_j}^k\}$, respectively.*

In self-paced learning, the goal is to jointly train the model parameter $\boldsymbol{w}$ and the latent weight variable $\boldsymbol{v}$ by minimizing

---

[3]Without ambiguity, we use $\ell_i(\boldsymbol{w})$ as the shorthand notations of $\ell\left(x_i, y_i; \boldsymbol{w}\right)$.

$$\operatorname*{argmin}_{\boldsymbol{w}\in\mathbb{R}^d, \boldsymbol{v}\in[0,1]^n} \mathcal{L}(\boldsymbol{w}, \boldsymbol{v}) := \sum_{j=1}^{m} \alpha_j \mathcal{R}_j(\boldsymbol{w}) + \sum_{i=1}^{n} \left[ v_i l_i(\boldsymbol{w}) + f(v_i, \lambda) \right], \tag{2}$$

where $f(v, \lambda)$ represents the SP-regularizer.

## 2.2 SP-regularizer

**Definition 2** (SP-regularizer [35]). *Suppose that $v$ is a weight variable, $\ell$ is the loss, and $\lambda$ is the age parameter. $f(v, \lambda)$ is called a self-paced regularizer, if*
*(i)$f(v, \lambda)$ is convex with respect to $v \in [0, 1]$;*
*(ii)$v^*(\ell, \lambda)$ is monotonically decreasing w.r.t. $\ell$, and holds $\lim_{\ell \to 0} v^*(\ell, \lambda) = 1, \lim_{\ell \to \infty} v^*(\ell, \lambda) = 0$;*
*(iii)$v^*(\ell, \lambda)$ is monotonically increasing w.r.t. $\lambda$, and holds $\lim_{\lambda \to \infty} v^*(\ell, \lambda) \leq 1, \lim_{\lambda \to 0} v^*(\ell, \lambda) = 0$,*
*where $v^*(\ell, \lambda) = \arg \min_{v \in [0,1]} v\ell + f(v, \lambda)$.*

The Definition 2 gives axiomatic definition of SP-regularizer. Some frequently utilized SP-regularizers include $f^H(v, \lambda) = -\lambda v$, $f^L(v, \lambda) = \lambda \left( \frac{1}{2} v^2 - v \right)$, $f^M(v, \lambda, \gamma) = \frac{\gamma^2}{v + \gamma/\lambda}$ and $f^{LOG}(v, \lambda, \alpha) = \frac{1}{\alpha} KL(1 + \alpha\lambda, v)$, which represents hard, linear, mixture, LOG SP-regularizer, respectively.

## 2.3 Biconvex Optimization

**Definition 3** (Biconvex Function). *A function $f : B \to \mathbb{R}$ on a biconvex set $B \subseteq \mathcal{X} \times \mathcal{Y}$ is called a biconvex function on $B$, if $f_x(\cdot) := f(x, \cdot) : B_x \to \mathbb{R}$ is a convex function on $B_x$ for every fixed $x \in \mathcal{X}$ and $f_y(\cdot) := f(\cdot, y) : B_y \to \mathbb{R}$ is a convex function on $B_y$ for every fixed $y \in \mathcal{Y}$.*

**Definition 4** (Partial Optimum). *Let $f : B \to \mathbb{R}$ be a given biconvex function and let $(x^*, y^*) \in B$. Then, $z^* = (x^*, y^*)$ is called a partial optimum of $f$ on $B$, if $f(x^*, y^*) \leq f(x, y^*) \forall x \in B_{y^*}$ and $f(x^*, y^*) \leq f(x^*, y) \forall y \in B_{x^*}$.*

Optimizing (2) leads to a biconvex optimization problem and is generally non-convex with fixed $\lambda$, in which a number of local minima exist and previous convex optimization tools can't achieve a promising effect [36]. It's reasonably believed that algorithms taking advantage of the biconvex structure are more efficient in the corresponding setting. For frequently used one, ACS (*c.f.* Algorithm 1) is presented to optimize $x$ and $y$ in $f(x, y)$ alternately until terminating condition is met.

---
**Algorithm 1** Alternate Convex Search (**ACS**)

**Require:** Dataset $X$ and $y$, age parameter $\lambda$.
1: Initialize $\boldsymbol{w}$.
2: **while** not converged **do**
3:     Update $\boldsymbol{v}^* = \operatorname{argmin}_{\boldsymbol{v}} \mathcal{L}(\boldsymbol{w}^*, \boldsymbol{v})$.
4:     Update $\boldsymbol{w}^* = \operatorname{argmin}_{\boldsymbol{w}} \mathcal{L}(\boldsymbol{w}, \boldsymbol{v}^*)$.
5: **end while**
**Ensure:** $\hat{w}$

---

**Remark 1.** *The order of the optimization subproblems in line 3 & 4 in Algorithm 1 can be permuted.*

**Theorem 1.** *[37] Let $\mathcal{X} \subseteq \mathbb{R}^n$ and $\mathcal{Y} \subseteq \mathbb{R}^m$ be closed sets and let $f : \mathcal{X} \times \mathcal{Y} \to \mathbb{R}$ be continuous. Let the sequence $\{z_i\}_{i \in \mathbb{N}_+}$ generated by ACS converges to $z^* \in \mathcal{X} \times \mathcal{Y}$. Then $z^*$ is a partial optimum.*

## 2.4 Theoretical Consistency

Researchers in earlier study [35] theoretically conducted the latent SPL loss (*a.k.a.*, implicit objective) and further proved that the SPL paradigm converges to the *stationary point* of the latent objective under some mild assumptions, which gives explanation to the robustness of the SPL [35, 38]. In this paper, we focus on the *partial optimum* of original SPL objective and result is given in Theorem 2.

**Theorem 2.** *Under the same assumptions in Theorem 2 of [38], the partial optimum of SPL objective consists with the stationary point of implicit SPL objective $G_\lambda$.*

Factoring in both Theorem 1 & 2, the ACS procedure (or its variations) used in SPL paradigm indeed finds the partial optimum of SPL objective, which unifies the two proposed analysis frameworks and provides more in-depth understanding to the intrinsic mechanism behind the SPL regime.

# 3 Age-Path Tracking

## 3.1 Objective Reformulation

For the convenience of derivation, we denote the set $I_{\mathcal{R}}$ or $\bar{I}_{\mathcal{R}}$ to be the set of indexes $j$ where $\mathcal{R}_j$ is differentiable or non-differentiable at $\boldsymbol{w}$, respectively. Similarly, we have $I_\ell$ and $\bar{I}_\ell$ w.r.t. $\ell_i$.

Moreover, the training of the SPL is essentially a process of adaptive sample selection, so we classify all the sample points in the training set into different sets $\mathcal{P} := \{\mathcal{E}, \mathcal{D}, \mathcal{M}, ...\}$ ac-

$$x_i \in \begin{cases} \mathcal{E}, & \text{if } l_i < \lambda \left(\text{or } \left(\dfrac{\lambda\gamma}{\lambda+\gamma}\right)^2 \text{ in mixture } f(v,\lambda)\right) \\ \mathcal{D}, & \text{if } l_i \geqslant \lambda \left(\text{or } \lambda^2 \text{ in mixture } f(v,\lambda)\right) \\ \mathcal{M}, & \text{if } \left(\dfrac{\lambda\gamma}{\lambda+\gamma}\right)^2 \leqslant l_i \leqslant \lambda^2 \text{ (only used in mixture } f(v,\lambda)) \end{cases}$$

cording to their confidence (or loss)[4]. Figure 3 illustrates a partition example when hard, linear or LOG SP-regularizer is used. Since subproblem in *line* 3 of Algorithm 1 always gives closed-form solutions in iterations[5], we can rewrite SPL optimization objective as (3), which is indeed equivalent to searching a partial optimum of (2).

$$\text{Compute } \hat{\boldsymbol{w}}, \text{ s.t. } \hat{\boldsymbol{w}} \in \arg \min_{\boldsymbol{w} \in \mathbb{R}^d} \sum_{j=1}^m \alpha_j \mathcal{R}_j(\boldsymbol{w}) + \sum_{J \in \mathcal{P}} \sum_{i \in J} v_i^*\left(l_i(\hat{\boldsymbol{w}}), \lambda\right) \cdot \ell_i(\boldsymbol{w}). \tag{3}$$

## 3.2 Piecewise Smooth Age-path

The KKT theorem [25] states that (3) holds iff

$$\boldsymbol{0} \in \sum_{j=1}^m \alpha_j \partial \mathcal{R}_j(\hat{\boldsymbol{w}}) + \sum_{J \in \mathcal{P}} \sum_{i \in J} v^*\left(l_i(\hat{\boldsymbol{w}}), \lambda\right) \cdot \partial \ell_i(\hat{\boldsymbol{w}}), \tag{4}$$

where $\partial(\cdot)$ denotes the subdifferential (set of all subgradients). In the $\mathbb{PC}^r$ setting, subgradient can be expressed explicitly by essentially active functions (*c.f.* Lemma 1).

**Definition 5** (Essentially Active Set). *Let $f : U \to \mathbb{R}$ be a $\mathbb{PC}^r$ function on the open set $U \in \mathbb{R}^n$ with a set of selection functions $\{f_i\}_{i \in I_f}$. For $\boldsymbol{x} \in U$, we call $I_f^a(\boldsymbol{x}) := \{i \in I_f : f(\boldsymbol{x}) = f_i(\boldsymbol{x})\}$ is the active set at $\boldsymbol{x}$, and $I_f^e(\boldsymbol{x}) := \left\{i \in I_f : \boldsymbol{x} \in \boldsymbol{cl}\left(\boldsymbol{int}\left(\{\boldsymbol{y} \in U : f(\boldsymbol{y}) = f_i(\boldsymbol{y})\}\right)\right)\right\}$ is the essentially active set at $\boldsymbol{x}$, where $\boldsymbol{cl}(\cdot)$ and $\boldsymbol{int}(\cdot)$ denote the closure and interior of a set.*

**Lemma 1.** *[39] Let $f : U \to \mathbb{R}$ be a $\mathbb{PC}^r$ function on an open set $U$ and $\bigcup_{i \in I_f}\{f_i\}$ is a set of selection functions of $f$, then $\partial f(\boldsymbol{x}) = \text{conv}(\bigcup_{i \in I_f^e(\boldsymbol{x})}\{f_i(x)\}) = \{\sum_{i \in I_f^e(\boldsymbol{x})} t_i \nabla f_i(\boldsymbol{x}) : \sum_{i \in I_f^e(\boldsymbol{x})} t_i = 1, \ t_i \geq 0\}$. Especially, if $f$ is differentiable at $\boldsymbol{x}$, $\partial f(\boldsymbol{x}) = \{\nabla f(\boldsymbol{x})\}$.*

**Assumption 2.** *We assume that $I_{\mathcal{R}_j}^a(\boldsymbol{x}) = I_{\mathcal{R}_j}^e(\boldsymbol{x}), I_{\ell_i}^a(\boldsymbol{x}) = I_{\ell_i}^e(\boldsymbol{x})$ holds for all $\boldsymbol{x}$ considered and all $\mathcal{R}_j, \ell_i$ in the following.*

We adopt a mild relaxation as shown in Assumption 2. Investigation [40] confirmed that it can be easily established in most practical scenarios. Without loss of generality, we suppose the following Assumption 3 also holds to further ease the notation burden.

**Assumption 3.** *We assume that $\mathcal{R}_j, \ell_i$ are non-differentiable at $\boldsymbol{x}$ with multiple active selection functions, where $j \in \{1, \ldots, m\}$, $i \in \{1, \ldots, n\}$.*

Therefore, the condition (4) can be rewritten in detail. Formally, there exists $\hat{\boldsymbol{t}}_{\mathcal{R}}$ and $\hat{\boldsymbol{t}}_\ell$ such that

---

[4]We only present the mainstream SP-regularizers here. The partition is similar in other cases.

[5]For example, we have $v_i^* = \begin{cases} -\ell_i/\lambda + 1, & \text{if } \ell_i < \lambda \\ 0, & \text{if } \ell_i \geq \lambda \end{cases}$ for linear $f(v, \lambda)$. More results are shown in [35].

$$\sum_{j=1}^{m} \sum_{k \in I_{\mathcal{R}_j}^a(\hat{\boldsymbol{w}})} \alpha_j \hat{t}_{\mathcal{R}_j}^k(\hat{\boldsymbol{w}}) \nabla D_{\mathcal{R}_j}^k(\hat{\boldsymbol{w}}) + \sum_{J \in \mathcal{P}} \sum_{i \in J} \sum_{k \in I_{\ell_i}^a(\hat{\boldsymbol{w}})} v_i^*\left(\ell_i(\hat{\boldsymbol{w}}), \lambda\right) \hat{t}_{\ell_i}^k(\hat{\boldsymbol{w}}) \nabla D_{\ell_i}^k(\hat{\boldsymbol{w}}) = \boldsymbol{0},$$

$$D_{\mathcal{R}_j}^k(\hat{\boldsymbol{w}}) - D_{\mathcal{R}_j}^{r_j}(\hat{\boldsymbol{w}}) = 0, \quad \forall k \in I_{\mathcal{R}_j}^a(\hat{\boldsymbol{w}}) \backslash \{r_j\}, \quad \forall j \in \bar{I}_{\mathcal{R}}$$

$$D_{\ell_i}^k(\hat{\boldsymbol{w}}) - D_{\ell_i}^{l_i}(\hat{\boldsymbol{w}}) = 0, \quad \forall k \in I_{\ell_i}^a(\hat{\boldsymbol{w}}) \backslash \{l_i\}, \quad \forall i \in \bar{I}_\ell \tag{5}$$

$$\sum_{k \in I_{\mathcal{R}_j}^a(\hat{\boldsymbol{w}})} \hat{t}_{\mathcal{R}_j}^k(\hat{\boldsymbol{w}}) - 1 = 0, \quad \hat{t}_{\mathcal{R}_j}^k(\hat{\boldsymbol{w}}) \geq 0, \quad 1 \leq j \leq m$$

$$\sum_{k \in I_{\ell_i}^a(\hat{\boldsymbol{w}})} \hat{t}_{\ell_i}^k(\hat{\boldsymbol{w}}) - 1 = 0, \quad \hat{t}_{\ell_i}^k(\hat{\boldsymbol{w}}) \geq 0, \quad 1 \leq i \leq n,$$

where $r_j, l_i$ is randomly selected from $I_{\mathcal{R}_j}^a$, $I_{\ell_i}^a$ and being fixed. The second and third equations in (5) describe the active sets while the last two equations describe the subgradients. When the partial optimum is on the smooth part, we denote the left side of equations (5) to be a $\mathbb{C}^1$ function $\mathcal{F}$, thus revealing that the solution path lies on the smooth manifold $\mathcal{F}(\boldsymbol{w}, \lambda, \boldsymbol{t}_{\mathcal{R}}, \boldsymbol{t}_\ell) = \boldsymbol{0}$. By the time it comes across the kink[6], we need to refresh the index partitions and update (5) to run next segment of path. WLOG, we postulate that the initial point is non-degenerate (*i.e.*, the $\boldsymbol{J}_{\boldsymbol{w}, \boldsymbol{t}_{\mathcal{R}}, \boldsymbol{t}_\ell}$ is invertible). By directly applying the implicit function theorem, the existence and uniqueness of a local $\mathbb{C}^1$ solution path $\left(\hat{\boldsymbol{w}}, \hat{\boldsymbol{t}}_{\mathcal{R}}, \hat{\boldsymbol{t}}_\ell\right)$ can be established over here. Drawing from the theory of differential geometry gives another intuitive understanding of age-path, which tells that the first equation in (5) indeed uses an analogue moving frame [41] to represent a smooth curve that consists of the smooth structure.

**Theorem 3.** *Given a partial optimum $(\hat{\boldsymbol{w}}, \boldsymbol{v}^*(\hat{\boldsymbol{w}}, \lambda))$ at $\lambda_0$, $\hat{\boldsymbol{t}}_{\mathcal{R}}, \hat{\boldsymbol{t}}_\ell$ in (5) can be solved from $\mathcal{F}\left(\hat{\boldsymbol{w}}, \lambda_0, \hat{\boldsymbol{t}}_{\mathcal{R}}, \hat{\boldsymbol{t}}_\ell\right) = \boldsymbol{0}$. If the Jacobian $\boldsymbol{J}_{\boldsymbol{w}, \boldsymbol{t}_{\mathcal{R}}, \boldsymbol{t}_\ell}$ is invertible at $\left(\hat{\boldsymbol{w}}, \hat{\boldsymbol{t}}_{\mathcal{R}}, \hat{\boldsymbol{t}}_\ell\right)$, then in an open neighborhood of $\lambda_0$, $\left(\hat{\boldsymbol{w}}, \hat{\boldsymbol{t}}_{\mathcal{R}}, \hat{\boldsymbol{t}}_\ell\right)$ is a $\mathbb{C}^1$ function w.r.t. $\lambda$ and fits the ODEs*

$$\frac{d \begin{pmatrix} \hat{\boldsymbol{w}} \\ \hat{\boldsymbol{t}}_{\mathcal{R}} \\ \hat{\boldsymbol{t}}_\ell \end{pmatrix}}{d\lambda} = -\boldsymbol{J}_{\boldsymbol{w}, \boldsymbol{t}_{\mathcal{R}}, \boldsymbol{t}_\ell}^{-1} \cdot \boldsymbol{J}_\lambda, \tag{6}$$

*in which the explicit expressions of $\boldsymbol{J}_{\boldsymbol{w}, \boldsymbol{t}_{\mathcal{R}}, \boldsymbol{t}_\ell}^{-1}$, $\boldsymbol{J}_\lambda$ are listed in Appendix A.*

**Corollary 1.** *If all the functions are smooth in a neighborhood of the initial point, then (6) can be simplified as $d\hat{\boldsymbol{w}}/d\lambda = -\boldsymbol{J}_{\boldsymbol{w}}^{-1} \cdot \boldsymbol{J}_\lambda$.*

**Remark 2.** *Our supplement parts in Appendix A present additional discussions.*

### 3.3 Critical Points

By solving the initial value problem (6) numerically with ODE solvers, the solution path regarding to $\lambda$ can be computed swiftly before any of $\mathcal{P}, I_{\mathcal{R}}$ or $I_\ell$ changes. We denote such point where the set changes a *critical point*, which can be divided into *turning point* or *jump point* on the basis of path's property at that point. To be more specific, the age-path is discontinuous at a jump point, while being continuous but non-differentiable at the turning point. This is also verified by Figure 2 and large quantity of numerical experiments.

At turning points, the operation of the algorithm is to update $\mathcal{P}, I_{\mathcal{R}}, I_\ell$ according to index violator(s) and move on to the next segment. At jump points, path is no longer continuous and warm-start[7] can be utilized to speed up the training procedure. The total number of critical points on the solution path is estimated at approximately $\mathcal{O}(|\mathcal{D} \cup \mathcal{M}|)$[8]. Consequently, we present a heuristic trick to figure out the type of a critical point with complexity $\mathcal{O}(d)$, so as to avoid excessive restarts. As a matter of fact, the solutions returned by the numerical ODE solver is continuous with the fixed set, despite it may actually passes a jump point. In this circumstance, the solutions returned by ODEs have deviated

---

[6]$\hat{\boldsymbol{t}}_{\bar{I}_{\mathcal{R}}}, \hat{\boldsymbol{t}}_{\bar{I}_\ell}$ hit the restriction bound in Lemma 1 or $I_\ell, \mathcal{P}$ are violated so that the entire structure changes.

[7]Reuse previous solutions. The subsequent calls to fit the model will not re-initialise parameters.

[8]Precisely speaking, it's related to interval length of $\lambda$, the nature of objective and the distribution of data.

---

**Algorithm 2** Generalized Age-path Algorithm (`GAGA`)

---

**Input**: Initial solution $\hat{\boldsymbol{w}}|_{\lambda_t=\lambda_{min}}$, $X$, $y$, $\lambda_{min}$ and $\lambda_{max}$.
**Output**: Age-Path $\hat{\boldsymbol{w}}(\lambda)$ on $[\lambda_{min}, \lambda_{max}]$.

 1: $\lambda_t \leftarrow \lambda_{min}$, set $\mathcal{P}, I_\mathcal{R}, I_\ell$ according to $\hat{\boldsymbol{w}}|_{\lambda_t}$.
 2: **while** $\lambda_t \leq \lambda_{max}$ **do**
 3:     Solve (6) and examine partition $\mathcal{P}, I_\mathcal{R}, I_\ell$ simultaneously.
 4:     **if** Partition $\mathcal{P}, I_\mathcal{R}, I_\ell$ was not met **then**
 5:        Update $\mathcal{P}, I_\mathcal{R}, I_\ell$ according to index violator(s).
 6:        Solve (6) with updated $\mathcal{P}, I_\mathcal{R}, I_\ell$.
 7:        **if** KKT conditions are not met **then**
 8:           Warm start at $\lambda_t + \delta$ (for a small $\delta > 0$).
 9:        **end if**
10:     **end if**
11: **end while**

---

from the ground truth partial optimum. Hence it's convenient that we can detect KKT conditions to monitor this behavior. This approach enjoys higher efficiency than detecting the partition conditions themselves, especially when the set partition is extraordinarily complex.

### 3.4 `GAGA` Algorithm

There has been extensive research in applied mathematics on numerical methods for solving ODEs, where the solver could automatically determine the step size of $\lambda$ when solving (6). In the tracking process, we compute the solutions with regard to $\lambda$. After detecting a new critical point, we need to reset $\mathcal{P}, I_\mathcal{R}, I_\ell$ at turning point while warm-start is required for jump point. The above procedure is repeated until we traverse the entire interval $[\lambda_{min}, \lambda_{max}]$. We show the detailed procedure in Algorithm 2. The main computational burden occurs in solving $\boldsymbol{J}^{-1}$ in (6) with an approximate complexity $\mathcal{O}(p^3)$ in general, where $p$ denotes the dimension of $\boldsymbol{J}$. Further promotion can be made via decomposition or utilizing the sparse representation of $\boldsymbol{J}$ on specific learning problems.

## 4 Practical Guides

In this section, we provide practical guides of using the `GAGA` to solve two important learning problems, *i.e.*, classic SVM and Lasso. The detailed steps of algorithms are displayed in Appendix C.

### 4.1 Support Vector Machines

Support vector machine (SVM) [33] has attracted much attention from researchers in the areas of bioinformatics, computer vision and pattern recognition. Given the dataset $X$ and label $\boldsymbol{y}$, we focus on the classic support vector classification as

$$\min_{\boldsymbol{w}, b} \frac{1}{2}\|\boldsymbol{w}\|_\mathcal{H}^2 + \sum_{i=1}^n C \max\{0,\ 1 - y_i(\langle \phi(x_i), \boldsymbol{w}\rangle + b)\}, \tag{7}$$

where $\mathcal{H}$ is the reproducing kernel Hilbert space (RKHS) with the inner product $\langle \cdot \rangle$ and corresponding kernel function $\phi$. Seeing that (5) still holds in infinite dimensional $\mathcal{H}$, the above analyses can be directly applied here. We also utilize the *kernel trick* [42] to avoid involving the explicit expression of $\phi$. In consistent with the framework, we have $\ell_i = C \max\{0, g_i\}$ and $g_i = 1 - y_i(\langle \phi(x_i), \boldsymbol{w}\rangle + b)$. The $I_\ell$ and $\mathcal{P}$ are determined by $\boldsymbol{g}$, thus we merely need to refine the division of $\mathcal{E}$ as $\mathcal{E}_N = \{i \in \mathcal{E} : g_i < 0\}, \mathcal{E}_Z = \{i \in \mathcal{E} : g_i = 0\}$ and $\mathcal{E}_P = \{i \in \mathcal{E} : g_i > 0\}$, which gives $I_\ell = \mathcal{E}_N \cup \mathcal{E}_P \cup \mathcal{D}(\cup \mathcal{M})$. Afterwards, with some simplifications and denoting $\hat{\boldsymbol{\alpha}} = C\boldsymbol{v}^* \odot \hat{\boldsymbol{t}}$, we can obtain a simplified version of (5), from where the age-path can be equivalently calculated w.r.t. optimal $(\hat{\boldsymbol{\alpha}}, \hat{b})$.

**Proposition 1.** *When $\boldsymbol{\alpha}, b$ indicate a partial optimum, the dynamics of optimal $\boldsymbol{\alpha}, b$ in (7) w.r.t. $\lambda$ for the linear and mixture SP-regularizer are shown as[9]*

$$\frac{d\begin{pmatrix} \boldsymbol{\alpha}_{\mathcal{E}_Z} \\ \boldsymbol{\alpha}_{\mathcal{E}_P} \\ b \end{pmatrix}}{d\lambda} = \begin{pmatrix} -\boldsymbol{y}_{\mathcal{E}_Z}^T & -\boldsymbol{y}_{\mathcal{E}_P}^T & 0 \\ Q_{\mathcal{E}_Z\mathcal{E}_Z} & Q_{\mathcal{E}_Z\mathcal{E}_P} & \boldsymbol{y}_{\mathcal{E}_Z} \\ \frac{C^2}{\lambda}Q_{\mathcal{E}_P\mathcal{E}_Z} & \frac{C^2}{\lambda}Q_{\mathcal{E}_P\mathcal{E}_P} - I_{\mathcal{E}_P\mathcal{E}_P} & \frac{C^2}{\lambda}\boldsymbol{y}_{\mathcal{E}_P} \end{pmatrix}^{-1} \begin{pmatrix} 0 \\ \boldsymbol{0}_{\mathcal{E}_Z} \\ -\frac{C}{\lambda^2}\boldsymbol{\ell}_{\mathcal{E}_P} \end{pmatrix}, \quad (8)$$

$$\frac{d\begin{pmatrix} \boldsymbol{\alpha}_{\mathcal{E}_Z} \\ \boldsymbol{\alpha}_{\mathcal{M}} \\ b \end{pmatrix}}{d\lambda} = \begin{pmatrix} -\boldsymbol{y}_{\mathcal{E}_Z}^T & -\boldsymbol{y}_{\mathcal{M}}^T & 0 \\ Q_{\mathcal{E}_Z\mathcal{E}_Z} & Q_{\mathcal{E}_Z\mathcal{M}} & \boldsymbol{y}_{\mathcal{E}_Z} \\ \frac{C^2\gamma}{2}\tilde{Q}_{\mathcal{M}\mathcal{E}_Z} & \frac{C^2\gamma}{2}\tilde{Q}_{\mathcal{M}\mathcal{M}} - I_{\mathcal{M}\mathcal{M}} & \frac{C^2\gamma}{2}\tilde{\boldsymbol{y}}_{\mathcal{M}} \end{pmatrix}^{-1} \begin{pmatrix} 0 \\ \boldsymbol{0}_{\mathcal{E}_Z} \\ -\frac{C\gamma}{\lambda^2}\boldsymbol{1}_{\mathcal{M}} \end{pmatrix}, \quad (9)$$

*respectively, where $\tilde{Q}_{\mathcal{M}\mathcal{E}_Z} = Diag\{\boldsymbol{\ell}_{\mathcal{M}}^{-\frac{3}{2}}\}Q_{\mathcal{M}\mathcal{E}_Z}, \tilde{Q}_{\mathcal{M}\mathcal{M}} = Diag\{\boldsymbol{\ell}_{\mathcal{M}}^{-\frac{3}{2}}\}Q_{\mathcal{M}\mathcal{M}}, \tilde{\boldsymbol{y}}_{\mathcal{M}} = \boldsymbol{\ell}_{\mathcal{M}}^{-\frac{3}{2}} \odot \boldsymbol{y}_{\mathcal{M}}$. Other components are constant as $\boldsymbol{\alpha}_{\mathcal{E}_N} = \boldsymbol{0}_{\mathcal{E}_N}, \boldsymbol{\alpha}_{\mathcal{D}} = \boldsymbol{0}_{\mathcal{D}}$. Only for mixture regularizer, $\boldsymbol{\alpha}_{\mathcal{E}_P} = \boldsymbol{1}_{\mathcal{E}_P}$.*

**Critical Point.** We track $\boldsymbol{g}$ along the path. The critical point is sparked off by any set in $\mathcal{E}_N, \mathcal{E}_Z, \mathcal{E}_P, \mathcal{D}(,\mathcal{M})$ changes.

### 4.2 Lasso

Lasso [34] uses a sparsity based regularization term that can produce sparse solutions. Given the dataset $X$ and label $\boldsymbol{y}$, the Lasso regression is stated as

$$\min_{\boldsymbol{w} \in \mathbb{R}^d} \frac{1}{2n}\|X\boldsymbol{w} - \boldsymbol{y}\|^2 + \alpha\|\boldsymbol{w}\|_1. \quad (10)$$

We expand $\|\boldsymbol{w}\|_1 = \sum_{j=1}^d |w_j|$ and treat $|w_j|$ as $\mathcal{R}_j$ in (5), hence the $I_{\mathcal{R}} = \{1 \leq j \leq d : w_j \neq 0\}$. We denote the set of active or inactive functions (components) by $\mathcal{A} = I_{\mathcal{R}}, \bar{\mathcal{A}} = \bar{I}_{\mathcal{R}}$, respectively. In view of the fact that $\partial|w_j|$ removes $\boldsymbol{t}_{\mathcal{R}_j}$ from the equations in (5) for $j \in \mathcal{A}$, we only pay attention to the $\mathcal{A}$ part w.r.t. the $(\boldsymbol{w}_{\mathcal{A}}, \lambda)$. The $\boldsymbol{\ell}$ is defined as $\frac{1}{2n}(X\boldsymbol{w} - \boldsymbol{y})^2$ in the following.

**Proposition 2.** *When $(\boldsymbol{w}, \boldsymbol{v}^*(\boldsymbol{w}, \lambda))$ is a partial optimum, the dynamics of optimal $\boldsymbol{w}$ in (10) w.r.t. $\lambda$ for the linear and mixture SP-regularizer are described as*

$$\frac{d\boldsymbol{w}_{\mathcal{A}}}{d\lambda} = -\frac{\sqrt{2n}}{\lambda^2}\left(X_{\mathcal{A}\mathcal{E}}^T Diag\left\{\boldsymbol{1}_{\mathcal{E}} - \frac{3}{\lambda}\boldsymbol{\ell}_{\mathcal{E}}\right\}X_{\mathcal{E}\mathcal{A}}\right)^{-1}X_{\mathcal{A}\mathcal{E}}^T\boldsymbol{\ell}_{\mathcal{E}}^{\frac{3}{2}}, \quad (11)$$

$$\frac{d\boldsymbol{w}_{\mathcal{A}}}{d\lambda} = -\frac{\sqrt{2n}\gamma}{\lambda^2}\left(X_{\mathcal{A}\mathcal{E}\cup\mathcal{M}}^T\tilde{X}_{\mathcal{E}\cup\mathcal{M}\mathcal{A}}\right)^{-1}X_{\mathcal{A}\mathcal{E}\cup\mathcal{M}}^T\begin{pmatrix} \boldsymbol{0}_{\mathcal{E}} \\ \boldsymbol{\ell}_{\mathcal{M}} \end{pmatrix}, \quad (12)$$

*respectively, where $\tilde{X}_{\mathcal{E}\cup\mathcal{M}\mathcal{A}} = \begin{pmatrix} X_{\mathcal{E}\mathcal{A}} \\ -\frac{\gamma}{\lambda}X_{\mathcal{M}\mathcal{A}} \end{pmatrix}$ and $\boldsymbol{w}_{\bar{\mathcal{A}}} = \boldsymbol{0}_{\bar{\mathcal{A}}}$.*

**Critical Point.** The critical point is encountered when $\mathcal{A}$ or $\mathcal{P}$ changes.

## 5 Experimental Evaluation

We present the empirical results of the proposed GAGA on two tasks: SVM for binary classification and Lasso for robust regression in the noisy environment. The results demonstrate that our approach outperforms existing SPL implementations on both tasks.

**Baselines.** GAGA is compared against three baseline methods: 1) **Original** learning model without SPL regime. 2) **ACS** [20] performs sequential search of $\lambda$, which is the most commonly used algorithm in SPL implementations. 3) **MOSPL** [29] is a state-of-the-art age-path approach that using the multi objective optimization, in which the solution is derived with the age parameter $\lambda$ implicitly.

---

[9]Notations such as $\boldsymbol{\ell}_{\mathcal{M}}^{-\frac{3}{2}}$ for vectors represent the element-wise operation in this section.

| Dataset | Source | Samples | Dimensions | Task |
|---------|--------|---------|------------|------|
| mfeat-pixel | UCI [43] | 2000 | 240 | |
| pendigits | UCI | 3498 | 16 | C |
| hiva agnostic | OpenML | 4230 | 1620 | |
| music | OpenML [44] | 1060 | 117 | |
| cadata | UCI | 20640 | 8 | |
| delta elevators | OpenML | 9517 | 8 | |
| houses | OpenML | 22600 | 8 | R |
| ailerons | OpenML | 13750 | 41 | |
| elevator | OpenML | 16600 | 18 | |

Table 1: Datasets description in our experiments. The C=Classification, R=Regression.

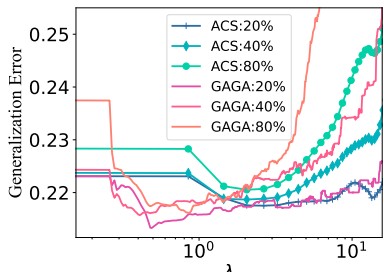

Figure 4: Robustness to noise. This figure shows the learning curve under different noise ratios, which confirms that the GAGA is more robust when in the setting of relatively high noise.

| Dataset | Parameter | | Competing Methods | | | Ours | Restarting times |
|---------|-----------|---|-------------------|---|---|------|------------------|
| | $C, \gamma_\kappa$ | $\alpha$ | *Original* | *ACS* | *MOSPL* | GAGA | |
| mfeat-pixel† | 1.00, 0.50 | – | 0.959±0.037 | 0.976±0.015 | 0.978±0.021 | **0.986**±0.016 | 23 |
| mfeat-pixel‡ | 1.00, 0.50 | – | 0.945±0.025 | 0.947±0.031 | 0.960±0.027 | **0.983**±0.013 | 25 |
| hiva agnostic† | 1.00, 0.50 | – | 0.868±0.027 | 0.941±0.009 | 0.946±0.0137 | **0.960**±0.004 | 8 |
| pendigits† | 1.00, 1.00 | – | 0.924±0.069 | 0.960±0.005 | 0.962±0.048 | **0.971**±0.046 | 10 |
| pendigits‡ | 1.00, 0.20 | – | 0.931±0.045 | 0.942±0.089 | 0.940±0.088 | **0.944**±0.089 | 8 |
| elevator† | – | 2e-3 | 0.146±0.011 | 0.144±0.012 | 0.144±0.020 | **0.143**±0.012 | 3 |
| ailerons† | – | 6e-3 | 0.674±0.071 | 0.492±0.006 | 0.491±0.041 | **0.489**±0.009 | 16 |
| music† | – | 5e-3 | 0.325±0.009 | 0.219±0.018 | 0.215±0.012 | **0.206**±0.013 | 123 |
| delta elevators† | – | 5e-3 | 0.783±0.153 | 0.724±0.138 | 0.679±0.057 | **0.634**±0.184 | 4 |
| houses† | – | 5e-3 | 0.213±0.013 | 0.209±0.010 | 0.205±0.231 | **0.201**±0.146 | 4 |

Table 2: Average results with the standard deviation in 20 runs on different datasets using the *linear SP-regularizer*. The top results in each row are in boldface.

| Dataset | Parameter | | | Competing Methods | | | Ours | Restarting times |
|---------|-----------|---|---|-------------------|---|---|------|------------------|
| | $\gamma$ | $C, \gamma_\kappa$ | $\alpha$ | *Original* | *ACS* | *MOSPL* | GAGA | |
| mfeat-pixel† | 0.20 | 1.00, 1.00 | – | 0.959±0.037 | 0.963±0.038 | 0.968±0.037 | **0.973**±0.040 | 12 |
| mfeat-pixel‡ | 0.50 | 0.20, 1.00 | – | 0.945±0.025 | 0.962±0.024 | 0.970±0.027 | **0.977**±0.015 | 10 |
| hiva agnostic† | 0.50 | 1.00, 1.00 | – | 0.868±0.027 | 0.946±0.004 | 0.949±0.019 | **0.957**±0.007 | 10 |
| pendigits† | 0.50 | 2.00, 1.00 | – | 0.924±0.069 | 0.956±0.062 | 0.957±0.071 | **0.962**±0.083 | 32 |
| pendigits‡ | 0.20 | 1.00, 1.00 | – | 0.931±0.045 | 0.940±0.088 | 0.942±0.089 | **0.944**±0.088 | 30 |
| cadata† | 1.00 | –,– | 5e-3 | 0.798±0.039 | 0.782±0.042 | 0.754±0.084 | **0.748**±0.010 | 13 |
| ailerons† | 0.50 | –,– | 5e-3 | 0.674±0.071 | 0.452±0.057 | 0.433±0.083 | **0.422**±0.090 | 14 |
| music† | 0.50 | –,– | 6e-3 | 0.325±0.009 | 0.218±0.008 | 0.216±0.021 | **0.213**±0.027 | 110 |
| delta elevators† | 0.50 | –,– | 5e-3 | 0.783±0.153 | 0.663±0.074 | 0.650±0.029 | **0.595**±0.132 | 12 |
| houses† | 0.50 | –,– | 5e-3 | 0.213±0.013 | 0.146±0.012 | 0.144±0.027 | **0.142**±0.012 | 8 |

Table 3: Average results with the standard deviation in 20 runs on different datasets using the *mixture SP-regularizer*. The top results in each row are in boldface.

**Datasets.** The Table 1 summarizes the datasets information. As universally known that SPL enjoys robustness in noisy environments, we impose 30% of noises into the real-world datasets. In particular, we generate noises by turning normal samples into poisoning ones by flipping their labels [45, 46] for classification tasks. For regression problem, noises are generated by the similar distribution of the training data as performed in [47].

**Experiment Setting.** In experiments, we first verify the performance of GAGA and traditional ACS algorithm under different noise intensity to reflect the robustness of GAGA. We further study the generalization performance of GAGA with competing methods, so as to show its ability to select optimal model during the learning process. Meanwhile, we also evaluate the running efficiency between GAGA and existing SPL implementations in different settings, which examines the speedup of GAGA as well as its practicability. Finally, we count the number of restarts and different types of

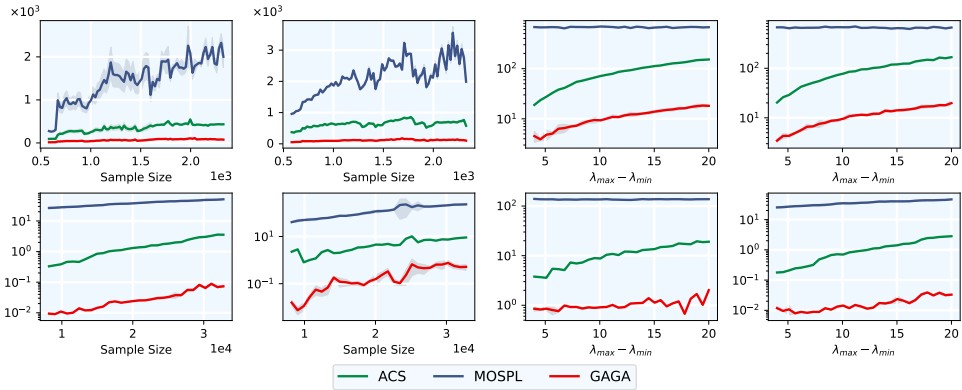

Figure 5: The study of efficiency comparison. $y$-axis denotes the average running time (in seconds) with 20 runs. The interval $[\lambda_{min}, \lambda_{max}]$ refers to the predefined searching area.

critical points when using `GAGA`, to investigate its ability to address critical points. For SVM, we use the Gaussian kernel $K(x_1, x_2) = \exp(-\gamma_\kappa \|x_1 - x_2\|^2)$. More details can be found in Appendix D.

**Results.** Figure 4 illustrates that conventional ACS fails to reduce the generalization error due to heavy noises fed to the model at a large age (*i.e.*, overfits the dataset), while `GAGA` makes up for this shortcoming by selecting the optimal model that merely learns the trust-worthy samples during the continuous learning process. Table 2 and 3 demonstrate an overall performance enhancing in `GAGA` than competing algorithms. The '†' in tables denotes 30% of artificial noise, while '‡' represents 20%. Note that performances are measured by accuracy and generalization error for classification and regression, respectively. The results guarantee that `GAGA` outperforms the state-of-the-art approaches in SPL under different circumstances. Figure 5 shows that `GAGA` also enjoys a high computational efficiency by changing the sample size as well as the predefined age sequence, emphasizing the potentials of utilizing `GAGA` in practice. The number of different types of critical points on some datasets is given in Figure 6. Corresponding restarting times can be found in Table 2 and 3, hence indicate that `GAGA` is capable of identifying different types of critical points and uses the heuristic trick to avoids restarts at massive turning points.

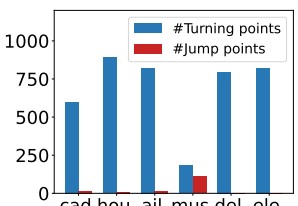

Figure 6: Histogram illustrating the number of different types of critical points. Names of datasets are shortened into the first 3 letters.

**Additional Experiments in Appendix D.** We further demonstrate the ability of `GAGA` to address the relatively large sample size and present more histograms. In addition, we apply `GAGA` to the logistic regression [33] for classification. We also verify that conventional ACS indeed tracks an approximation path of partial optimum in experiments, which provides a more in-depth understanding towards SPL and the performance promotion brought by `GAGA`. We also conduct a comparative study to the state-of-the-art robust model for SVMs [48, 49, 50] and Lasso [51] besides the SPL domain.

## 6    Conclusion

In this paper, we connect the SPL paradigm to the partial optimum for the first time. Using this idea, we propose the first *exact* age-path algorithm able to tackle *general* SP-regularizer, namely `GAGA`. Experimental results demonstrate `GAGA` outperforms traditional SPL paradigm and the state-of-the-art age-path approach in many aspects, especially in the highly noisy environment. We further build the relationship between our framework and existing theoretical analyses on SPL regime, which provides more in-depth understanding towards the principle behind SPL.

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
