# Appendix

## A    Supplementary Notes for Section 3

In this section, we present some additional discussions and theoretical results for Section 3.

### A.1    Explicit Expression

The goal of this section is to obtain precise and explicit expressions of the Jacobian matrices in (6). Firstly, the $J_{w,t_\mathcal{R},t_\ell}^{-1}$ has the form of

$$J_{w,t_\mathcal{R},t_\ell}^{-1} = \left( \begin{array}{cc} \tilde{F}_w & \tilde{F}_t \\ \left(\begin{array}{c} \tilde{D} \\ O \end{array}\right) & \left(\begin{array}{c} O \\ \tilde{I} \end{array}\right) \end{array} \right)^{-1}, \tag{13}$$

in which

$$\tilde{F}_w = \sum_{j=1}^m \sum_{k \in I_{\mathcal{R}_j}^a(\hat{w})} \alpha_j \hat{t}_{\mathcal{R}_j}^k(\hat{w}) \nabla^2 D_{\mathcal{R}_j}^k(\hat{w}) + \sum_{J \in \mathcal{P}} \sum_{i \in J} \sum_{k \in I_{\ell_i}^a(\hat{w})} v_i^* \left(\ell_i(\hat{w}), \lambda\right) \hat{t}_{\ell_i}^k(\hat{w}) \nabla^2 D_{\ell_i}^k(\hat{w}),$$

$$\tilde{F}_t = \left( \begin{array}{cc} \texttt{+\!\!\!+}_{1 \le j \le m, k \in I_{\mathcal{R}_j}^a} \nabla D_{\mathcal{R}_j}^k(\hat{w}) & \texttt{+\!\!\!+}_{J \in \mathcal{P}, j \in J, k \in I_{\ell_i}^a} \alpha_j \nabla D_{\ell_j}^k(\hat{w}) \end{array} \right),$$

$$\tilde{D} = \left( \begin{array}{c} \left( \texttt{+\!\!\!+}_{1 \le j \le m, k \in I_{\mathcal{R}_j}^a \setminus \{r_j\}} \nabla D_{\mathcal{R}_j}^k(\hat{w}) - \nabla D_{\mathcal{R}_j}^{r_j}(\hat{w}) \right)^T \\ \left( \texttt{+\!\!\!+}_{J \in \mathcal{P}, j \in J, k \in I_{\ell_i}^a \setminus \{l_i\}} \nabla D_{\ell_i}^k(\hat{w}) - \nabla D_{\ell_i}^{l_i}(\hat{w}) \right)^T \end{array} \right),$$

$$\tilde{I} = \left( \begin{array}{c} \left( \texttt{+\!\!\!+}_{i=1}^n \left( \begin{array}{c} \mathbf{1}_{I_{\mathcal{R}_j}^a} \\ \mathbf{0}_{\bar{I}_{\mathcal{R}_j}^a} \end{array} \right) \right)^T \\ \left( \texttt{+\!\!\!+}_{J \in \mathcal{P}, j \in J} \left( \begin{array}{c} \mathbf{1}_{I_{\ell_i}^a} \\ \mathbf{0}_{\bar{I}_{\ell_i}^a} \end{array} \right) \right)^T \end{array} \right).$$

The symbol $\texttt{+\!\!\!+}$ denotes column matrix concatenation, following the convention in Haskell Language. Similarly, $J_\lambda$ is given by

$$J_\lambda = \left( \begin{array}{c} \tilde{F}_\lambda \\ \mathbf{0} \end{array} \right), \tag{14}$$

where

$$\tilde{F}_\lambda = \sum_{J \in \mathcal{P}} \sum_{i \in J} \sum_{k \in I_{\ell_i}^a(\hat{w})} \frac{\partial v_i^* \left(\ell_i(\hat{w}), \lambda\right)}{\partial \lambda} \hat{t}_{\ell_i}^k(\hat{w}) \nabla D_{\ell_i}^k(\hat{w}).$$

As an outcome, (6) can be explicitly expressed via

$$\frac{d \left( \begin{array}{c} \hat{w} \\ \hat{t}_\mathcal{R} \\ \hat{t}_\ell \end{array} \right)}{d\lambda} = \left( \begin{array}{cc} \tilde{F}_w & \tilde{F}_t \\ \left(\begin{array}{c} \tilde{D} \\ O \end{array}\right) & \left(\begin{array}{c} O \\ \tilde{I} \end{array}\right) \end{array} \right)^{-1} \left( \begin{array}{c} \tilde{F}_\lambda \\ \mathbf{0} \end{array} \right). \tag{15}$$

### A.2    On the Numerical ODEs Solving

**Matrix Inversion.**    Here we give some relative discussions to support the assumption in Section 3.2 that $J_{w,t_\mathcal{R},t_\ell}$ ($J$ for short) is non-singular. First note that $J$ is singular if and only if the value of its determinant $|J|$ is zero, where $|J|$ is indeed a polynomial w.r.t. the uncertain elements in $J$. Denote the number of these unknown components as $q$, then the probability that $J$ is singular can be somewhat equivalently seen as the measure of the hypersurface $\mathcal{S} = \{x \in \mathbb{R}^q : |J(x)| = 0\}$ in $\mathbb{R}^q$. For the polynomial function $|J|$, it's easy to prove that $\mathcal{S}$ is a zero-measured set in $\mathbb{R}^q$, which

indicates the probability of $\boldsymbol{J}$ being non-invertible is zero. This result shows the non-singularity assumption fits the common situation in practice. Secondly, there are some general cases that $\boldsymbol{J}$ is guaranteed to be invertible. We refer some of these claims to [40]. Moreover, during the extensive empirical studies, none of the singular $\boldsymbol{J}$ is observed, which again validates the rationality of the assumption.

**Robustness.** In practice, we avoid directly computing the inverse of $\boldsymbol{J}$ by the consideration of robustness. Instead, we adopt the Moore-Penrose inverse [52] in implementation. The Moore-Penrose inverse exists for any matrix $X$ even if the matrix owns singularity, which guarantees our algorithm to be robust.

**Complexity.** Our approach utilizes the singular value decomposition (SVD) [52] when solving the Moore-Penrose inverse in (15), which is demonstrated to be a state-of-the-art technique via a computationally simple and precise way [53]. Consequently, the computational efficiency as well as the accuracy of our algorithm is guaranteed.

**Stability.** Our algorithm uses ODE solvers from the LSODE package [54] when solving the initial value problem (6). The solver will automatically select a proper method to solve different initial value problems which guarantees the general performance of our algorithm. Especially, when the problem tends to be unstable, the solver adopts the backward differentiation formula (BDF) method [55] to avoid extremely small step sizes while preserving the stability and accuracy of output solutions.

## B    Proofs

In this section, we give complete proofs to all the theorems and properties stated in the main article.

### B.1    Proof of Theorem 2

Prior to the proof, we first review the relevant background about latent SPL loss [38]. Regarding the unconstrained learning problem (1), its latent SPL objective is defined as[10]

$$G_\lambda(\boldsymbol{w}) := \sum_{j=1}^{m} \alpha_j \mathcal{R}_j(\boldsymbol{w}) + \sum_{i=1}^{n} F_\lambda\left(\ell_i(\boldsymbol{w})\right), \tag{16}$$

where $F_\lambda(\ell) = \int_0^\ell v_\lambda^*(\tau)\,d\tau$.

**Theorem 4.** *[38] In the SPL objective (2), suppose $\ell$ is bounded below, $\boldsymbol{w} \mapsto \ell(\cdot)$ is continuously differentiable, $v_\lambda^*(\cdot)$ is continuous, and $\sum_{j=1}^{m} \alpha_j \mathcal{R}_j$ is coercive and lower semi-continuous. Then for any initial parameter $\boldsymbol{w}^0$, every cluster point of the produced sequence $\{\boldsymbol{w}^k\}$, obtained by the ACS algorithm on solving (2), is a critical point of the implicit objective $G_\lambda$ (16).*

In Theorem 2, the relationship between the partial optimum of original SPL objective $\mathcal{L}$ and the critical point of implicit SPL objective $G_\lambda$ is constructed. Its proof is given as follows.

*Proof.* On one hand,

$$\boldsymbol{w}_0 \text{ is a critical point of } G_\lambda \iff 0 \in \partial G_\lambda\left(\boldsymbol{w}_0\right) = \partial \sum_{j=1}^{m} \alpha_j \mathcal{R}_j(\boldsymbol{w}_0) + \sum_{i=1}^{N} \nabla F_\lambda\left(l_i(\boldsymbol{w}_0)\right)$$

$$= \sum_{j=1}^{m} \alpha_j \partial \mathcal{R}_j(\boldsymbol{w}_0) + \sum_{i=1}^{N} v_\lambda^*\left(l_i(\boldsymbol{w}_0)\right) \cdot \nabla l_i(\boldsymbol{w}_0).$$

---

[10]Note that the objective function in (1) is indeed the same as what in [38] despite minor differences in notations, so we keep the manner in this paper for the sake of consistency.

On the other hand, assuming $(\boldsymbol{w}_0, \boldsymbol{v})$ is a partial optimum of the SPL objective $\mathcal{L}$, it's obvious that $v_i = v_\lambda^*(l_i(\boldsymbol{w}_0))$ (for $i = 1 \ldots N$), where we write $\boldsymbol{v} = \boldsymbol{v}_\lambda^*(\boldsymbol{w}_0)$ in short. Then

$$(\boldsymbol{w}_0, \boldsymbol{v}_\lambda^*(\boldsymbol{w}_0)) \text{ is a partial optimum of } \mathcal{L} \iff 0 \in \partial_{\boldsymbol{w}} \mathcal{L}(\boldsymbol{w}_0, \boldsymbol{v}_\lambda^*(\boldsymbol{w}_0); \lambda)$$
$$= \partial \sum_{j=1}^{m} \alpha_j \mathcal{R}_j(\boldsymbol{w}_0) + \sum_{i=1}^{N} v_\lambda^*(l_i(\boldsymbol{w}_0)) \cdot \nabla l_i(\boldsymbol{w}_0).$$

Combine the above two results we can conclude Theorem 2. $\qquad\square$

## B.2 Proof of Theorem 3

*Proof.* In Section 3.2, we have shown that any $(\boldsymbol{w}, \boldsymbol{v}^*(\boldsymbol{\ell}(\boldsymbol{w}), \lambda))$ is a partial optimum iff there exists $\boldsymbol{t}_{\mathcal{R}}, \boldsymbol{t}_\ell$ such that (5) holds. Given a certain partial optimum $(\hat{\boldsymbol{w}}, \boldsymbol{v}^*(\boldsymbol{\ell}(\hat{\boldsymbol{w}}), \lambda))$, solving the corresponding $\hat{\boldsymbol{t}}_{\mathcal{R}}, \hat{\boldsymbol{t}}_\ell$ is indeed calculating the linear equations

$$\sum_{j=1}^{m} \sum_{k \in I_{\mathcal{R}_j}^a(\hat{\boldsymbol{w}})} \alpha_j \hat{t}_{\mathcal{R}_j}^k(\hat{\boldsymbol{w}}) \nabla D_{\mathcal{R}_j}^k(\hat{\boldsymbol{w}}) + \sum_{J \in \mathcal{P}} \sum_{i \in J} \sum_{k \in I_{\ell_i}^a(\hat{\boldsymbol{w}})} v_i^*(l_i(\hat{\boldsymbol{w}}), \lambda) \hat{t}_{\ell_i}^k(\hat{\boldsymbol{w}}) \nabla D_{\ell_i}^k(\hat{\boldsymbol{w}}) = \boldsymbol{0},$$

$$\sum_{k \in I_{\mathcal{R}_j}^a(\hat{\boldsymbol{w}})} \hat{t}_{\mathcal{R}_j}^k(\hat{\boldsymbol{w}}) - 1 = 0, \quad \hat{t}_{\mathcal{R}_j}^k(\hat{\boldsymbol{w}}) \geq 0, \quad 1 \leq j \leq m \tag{17}$$

$$\sum_{k \in I_{\ell_i}^a(\hat{\boldsymbol{w}})} \hat{t}_{\ell_i}^k(\hat{\boldsymbol{w}}) - 1 = 0, \quad \hat{t}_{\ell_i}^k(\hat{\boldsymbol{w}}) \geq 0, \quad 1 \leq i \leq n.$$

On a non-critical point, suppose we've obtained a partial optimum $(\hat{\boldsymbol{w}}, \boldsymbol{v}^*(\boldsymbol{\ell}(\hat{\boldsymbol{w}}), \lambda))$ at $\lambda$. Now a critical point is triggered by either of two conditions: **1)** Partition $\mathcal{P}$ changes. This means the value of some $\ell_i$ lie on the boundary between two distinct sets in $\mathcal{P}$. **2)** One of $I_{\mathcal{R}}, I_\ell$ changes. This indicates the existence of some $\mathcal{R}_i$ ($i \in \bar{I}_{\mathcal{R}}(I_{\mathcal{R}})$) becomes (non-)differentiable at $\hat{\boldsymbol{w}}$, or some $\ell_j$ ($j \in \bar{I}_\ell(I_\ell)$) becomes (non-)differentiable at $\hat{\boldsymbol{w}}$. The latter can be detected by the value of $t_i$. For example, assume that $i \in I_\ell$ holds along a segment of the path, i.e., there exists $k$ such that $t_{\ell_i}^k = 1$, while $t_{\ell_i}^{\tilde{k}} = 1$ holds for all $\tilde{k} \neq k$. At the kink, $\ell_i$ changes into non-differentiable. As a result, the value of $t_{\ell_i}^k$ will decrease from 1, since some other selection functions turn to essentially active status. Altogether, at the optimal $\hat{\boldsymbol{w}}$, all the inequalities in (5) are *strict*. In case that $\mathcal{F}(\boldsymbol{w}, \lambda, \boldsymbol{t}_{\mathcal{R}}, \boldsymbol{t}_\ell) = \boldsymbol{0}$ deduces a continuous solution path passing $(\hat{\boldsymbol{w}}, \hat{\boldsymbol{t}}_{\mathcal{R}}, \hat{\boldsymbol{t}}_\ell)$, the (5) will be maintained along the path until the next critical point occurs.

Denote the Jacobin of $\mathcal{F}$ w.r.t. $(\boldsymbol{w}, \boldsymbol{t}_{\mathcal{R}}, \boldsymbol{t}_\ell)$, $\lambda$ as $\boldsymbol{J}_{\boldsymbol{w}, \boldsymbol{t}_{\mathcal{R}}, \boldsymbol{t}_\ell}$, $\boldsymbol{J}_\lambda$, respectively. Following our setting in Section 3.2, $\mathcal{F}$ is $\mathbb{C}^1$ and $\boldsymbol{J}_{\boldsymbol{w}, \boldsymbol{t}_{\mathcal{R}}, \boldsymbol{t}_\ell}$ is invertible at the initial point. In this condition, the implicit function theorem directly indicates the existence and uniqueness of a local $\mathbb{C}^1$ solution path of optimal $(\hat{\boldsymbol{w}}, \hat{\boldsymbol{t}}_{\mathcal{R}}, \hat{\boldsymbol{t}}_\ell)$ w.r.t. $\lambda$, started from the initial point. Furthermore, the theorem also guarantees that (6) is valid along the path. Owing to the $\mathbb{C}^1$ function $\mathcal{F}$, the right side of (6) is continuous w.r.t. $\lambda$. Therefore, the Picard–Lindelöf theorem [56] straightforwardly proves that the solution of (6) is unique and can be extended to the nearest boundary of $\lambda$ (*i.e.*, a new critical point appears). $\qquad\square$

**Geometric Intuition.** There exists a geometric understanding towards (5) either. Rewriting the first equation in (5) by the differentiability of each function gives

$$-\sum_{j \in I_{\mathcal{R}}} \alpha_j \nabla \mathcal{R}_j - \sum_{i \in I_\ell} v^*(\ell_i, \lambda) \nabla \ell_i = \sum_{j \in \bar{I}_{\mathcal{R}}} \sum_{k \in I_{\mathcal{R}_j}^a} \hat{t}_{\mathcal{R}_j}^k \alpha_j \nabla D_{\mathcal{R}_j}^k + \sum_{i \in \bar{I}_\ell} \sum_{k \in I_{\ell_i}^a} \hat{t}_{\ell_i}^k v^*(\ell_i, \lambda) \nabla D_{\ell_i}^k, \tag{18}$$

where $\hat{t}_{\mathcal{R}_j}^k, \hat{t}_{\ell_i}^k$ meet restrictions in (5) and all symbols are in terms of $\hat{\boldsymbol{w}}$. As $\lambda$ varies, the left side in (18) describes a smooth curve using the standard frame in $\mathbb{R}^d$, while the right side is actually sum of vectors chosen from a convex hull $\text{conv}\left(\bigcup_{k \in I_{\mathcal{R}_j}^a} \{\alpha_j \nabla \{D_{\mathcal{R}_j}^k\}\}\right)$ or $\text{conv}\left(\bigcup_{k \in I_{\ell_i}^a} \{v^*(\ell_i, \lambda) \nabla D_{\ell_i}^k\}\right)$ and can be viewed as the vector $\boldsymbol{1}$ under an analogue moving frame made up of these selected vectors, from the perspective of differential geometry. In other words, (18) indeed uses an analogue moving frame to re-depict a smooth curve.

It is worth adding that a recent work [40] shows a similar intuition. Specifically, under some assumptions the solution surface near a given point is a projection of certain smooth manifold even without the convexity assumption.

### B.3 Proof of Corollary 1

*Proof.* Suppose that $\mathcal{R}_j, \ell_i$ in (1) are all differentiable at the given $\hat{\boldsymbol{w}}$, then it yields $I_\mathcal{R} = I_\ell = \emptyset$. Consequently, (5) is degenerate into its first equation

$$\mathcal{F}(\hat{\boldsymbol{w}}, \lambda) = \sum_{j=1}^{m} \alpha_j \nabla \mathcal{R}_j(\hat{\boldsymbol{w}}) + \sum_{J \in \mathcal{P}} \sum_{i \in J} v_i^* \left(\ell_i(\hat{\boldsymbol{w}}), \lambda\right) \nabla \ell_i(\hat{\boldsymbol{w}}) = \mathbf{0}. \tag{19}$$

Hence applying Theorem 3 straightly gives our Corollary 1. $\qquad\square$

### B.4 Proof of Prop. 1

Here we only present the detailed proof for SVM with linear SP-regularizer, due to the fact that the proof for mixture SP-regularizer is almost the same as the former except some partition difference.

*Proof.* Given a partial optimum $((\hat{\boldsymbol{w}}, b) \boldsymbol{v}^* (\hat{\boldsymbol{w}}, b))$ at $\lambda$, (5) can be directly applied here with obvious simplifications. Mathematically, there exits $\hat{\boldsymbol{t}} = \left(\hat{t}_i\right)_{i=1}^{n}$ such that

$$\hat{\boldsymbol{w}} - \sum_{i=1}^{n} C v_i^* \hat{t}_i y_i \phi(x_i) = \mathbf{0},$$

$$\sum_{i=1}^{n} C v_i^* \hat{t}_i y_i = \mathbf{0}, \tag{20}$$

$$1 - y_i \left(\langle \phi(x_i), \hat{\boldsymbol{w}} \rangle + b \right) = 0, \quad i \in \mathcal{E}_Z,$$

$$\hat{\boldsymbol{t}}_{\mathcal{E}_N} = \mathbf{0}_{\mathcal{E}_N}, \quad \hat{\boldsymbol{t}}_{\mathcal{E}_P \cup \mathcal{D}} = \mathbf{1}_{\mathcal{E}_P \cup \mathcal{D}}, \quad \mathbf{0} \preceq \hat{\boldsymbol{t}}_{\mathcal{E}_Z} \preceq \mathbf{1}_{\mathcal{E}_Z},$$

where $\preceq$ denotes the element-wise comparison between vectors. Denote $\hat{\boldsymbol{\alpha}} = C \boldsymbol{v}^* \odot \hat{\boldsymbol{t}}$, then (20) can be equivalently converted to equations w.r.t. $(\hat{\boldsymbol{\alpha}}, b)$ as

$$\boldsymbol{y}^T \hat{\boldsymbol{\alpha}} = \mathbf{0},$$

$$\mathbf{1}_{\mathcal{E}_Z} - Q_{\mathcal{E}_Z} \hat{\boldsymbol{\alpha}} - \boldsymbol{y}_{\mathcal{E}_Z} \hat{b} = \mathbf{0},$$

$$\hat{\boldsymbol{\alpha}}_{\mathcal{E}_P} - C \boldsymbol{v}_{\mathcal{E}_P}^* = \mathbf{0}_{\mathcal{E}_P}, \tag{21}$$

$$\hat{\boldsymbol{\alpha}}_{\mathcal{E}_Z} - C \boldsymbol{v}_{\mathcal{E}_Z}^* \odot \hat{\boldsymbol{t}}_{\mathcal{E}_Z} = \mathbf{0}_{\mathcal{E}_Z},$$

$$\hat{\boldsymbol{\alpha}}_{\mathcal{E}_N \cup \mathcal{D}} = \mathbf{0}_{\mathcal{E}_N \cup \mathcal{D}},$$

where $Q = \boldsymbol{y}^T K \boldsymbol{y}$, $K = (k(x_i, x_j))_{1 \leq i,j \leq n}$ is the kernel matrix and $k$ is the kernel function. Then the optimal $\hat{\boldsymbol{w}} = \sum_{i=1}^{n} \hat{\alpha}_i y_i \phi(x_i)$, hence problem (7) is transformed into solving equations merely related to $(\hat{\boldsymbol{\alpha}}, b)$. Specifically, the decision function can be rewritten as $d(x) = \sum_{i=1}^{n} y_i \hat{\alpha}_i k(x_i, x) + b$.

Now supposed that $(\hat{\boldsymbol{\alpha}}, \hat{b})$ is not a critical point, then the fourth equation in (21) is actually an inequality constraint by changing the value of $\hat{\boldsymbol{t}}_{\mathcal{E}_Z}$ in $[0,1]^{|\mathcal{E}_Z|}$. As a result, (21) is only related to $(\hat{\boldsymbol{\alpha}}, \hat{b})$ and the left side of the first three equations accords with the function $\mathcal{F}$ in Theorem 3. Consequently, (8) can be derived using Theorem 3. In detail, denote

$$\mathcal{F}(\hat{\boldsymbol{\alpha}}_{\mathcal{E}_Z}, \hat{\boldsymbol{\alpha}}_{\mathcal{E}_P}, \hat{b}) = \begin{pmatrix} \boldsymbol{y}_{\mathcal{E}_Z \cup \mathcal{E}_P}^T \hat{\boldsymbol{\alpha}}_{\mathcal{E}_Z \cup \mathcal{E}_P} \\ \mathbf{1}_{\mathcal{E}_Z} - Q_{\mathcal{E}_Z} \hat{\boldsymbol{\alpha}}_{\mathcal{E}_Z \cup \mathcal{E}_P} - \boldsymbol{y}_{\mathcal{E}_Z} \hat{b} \\ \hat{\boldsymbol{\alpha}}_{\mathcal{E}_P} - C \boldsymbol{v}_{\mathcal{E}_P}^* \end{pmatrix}$$

$$= \begin{pmatrix} \boldsymbol{y}_{\mathcal{E}_Z \cup \mathcal{E}_P}^T \hat{\boldsymbol{\alpha}}_{\mathcal{E}_Z \cup \mathcal{E}_P} \\ \mathbf{1}_{\mathcal{E}_Z} - Q_{\mathcal{E}_Z} \hat{\boldsymbol{\alpha}}_{\mathcal{E}_Z \cup \mathcal{E}_P} - \boldsymbol{y}_{\mathcal{E}_Z} \hat{b} \\ \hat{\boldsymbol{\alpha}}_{\mathcal{E}_P} - C(\mathbf{1}_{\mathcal{E}_P} - C \frac{\mathbf{1}_{\mathcal{E}_P} - Q_{\mathcal{E}_P} \hat{\boldsymbol{\alpha}}_{\mathcal{E}_P \cup \mathcal{E}_Z} - \boldsymbol{y}_{\mathcal{E}_P} \hat{b}}{\lambda}), \end{pmatrix}$$

then the jacobian can be calculated as

$$\boldsymbol{J}_{\mathcal{F},(\hat{\boldsymbol{\alpha}}_{\mathcal{E}_Z},\hat{\boldsymbol{\alpha}}_{\mathcal{E}_P},b)} = - \begin{pmatrix} -\boldsymbol{y}_{\mathcal{E}_Z}^T & -\boldsymbol{y}_{\mathcal{E}_P}^T & 0 \\ Q_{\mathcal{E}_Z \mathcal{E}_Z} & Q_{\mathcal{E}_Z \mathcal{E}_P} & \boldsymbol{y}_{\mathcal{E}_Z} \\ \frac{C^2}{\lambda} Q_{\mathcal{E}_P \mathcal{E}_Z} & \frac{C^2}{\lambda} Q_{\mathcal{E}_P \mathcal{E}_P} - I_{\mathcal{E}_P \mathcal{E}_P} & \frac{C^2}{\lambda} \boldsymbol{y}_{\mathcal{E}_P} \end{pmatrix},$$

$$\boldsymbol{J}_{\mathcal{F},\lambda} = \begin{pmatrix} 0 \\ \boldsymbol{0}_{\mathcal{E}_Z} \\ -\frac{C}{\lambda^2} \boldsymbol{\ell}_{\mathcal{E}_P} \end{pmatrix}.$$

The implicit function theorem immediately indicates the following ODEs hold

$$\frac{d \begin{pmatrix} \boldsymbol{\alpha}_{\mathcal{E}_Z} \\ \boldsymbol{\alpha}_{\mathcal{E}_P} \\ b \end{pmatrix}}{d\lambda} = -\boldsymbol{J}_{\mathcal{F},(\hat{\boldsymbol{\alpha}}_{\mathcal{E}_Z},\hat{\boldsymbol{\alpha}}_{\mathcal{E}_P},b)}^{-1} \cdot \boldsymbol{J}_{\mathcal{F},\lambda}$$

$$= \begin{pmatrix} -\boldsymbol{y}_{\mathcal{E}_Z}^T & -\boldsymbol{y}_{\mathcal{E}_P}^T & 0 \\ Q_{\mathcal{E}_Z \mathcal{E}_Z} & Q_{\mathcal{E}_Z \mathcal{E}_P} & \boldsymbol{y}_{\mathcal{E}_Z} \\ \frac{C^2}{\lambda} Q_{\mathcal{E}_P \mathcal{E}_Z} & \frac{C^2}{\lambda} Q_{\mathcal{E}_P \mathcal{E}_P} - I_{\mathcal{E}_P \mathcal{E}_P} & \frac{C^2}{\lambda} \boldsymbol{y}_{\mathcal{E}_P} \end{pmatrix}^{-1} \begin{pmatrix} 0 \\ \boldsymbol{0}_{\mathcal{E}_Z} \\ -\frac{C}{\lambda^2} \boldsymbol{\ell}_{\mathcal{E}_P} \end{pmatrix}.$$

$\square$

## B.5 Proof of Prop. 2

*Proof.* Following discussions in Section 4.2, the objective (10) of Lasso is reformulated under the self-paced paradigm as

$$\text{Compute } \hat{\boldsymbol{w}}, \text{ s.t. } \hat{\boldsymbol{w}} \in \arg\min_{\boldsymbol{w}} \alpha \|\boldsymbol{w}\|_1 + \frac{1}{2n}\|\sqrt{V^*}(X\boldsymbol{w} - \boldsymbol{y})\|^2, \tag{22}$$

where $V^*$ denotes $Diag\{\boldsymbol{v}^*\}$ and $\sqrt{V^*}$ denotes $Diag\{\sqrt{\boldsymbol{v}^*}\}$. Given a partial optimum $\hat{\boldsymbol{w}}$ at $\lambda$, applying (5) to the objective (10) deduces

$$\frac{1}{n} X_{\mathcal{A}}^T V^* (X\hat{\boldsymbol{w}} - \boldsymbol{y}) + \alpha \cdot \mathbf{sgn}(\hat{\boldsymbol{w}}_{\mathcal{A}}) = \boldsymbol{0},$$

$$\frac{1}{n} X_{\bar{\mathcal{A}}}^T V^* (X\hat{\boldsymbol{w}} - \boldsymbol{y}) + \alpha \cdot \hat{\boldsymbol{t}}_{\bar{\mathcal{A}}} = \boldsymbol{0}, \tag{23}$$

where $-\mathbf{1}_{\bar{\mathcal{A}}} \preceq \hat{\boldsymbol{t}}_{\bar{\mathcal{A}}} \preceq \mathbf{1}_{\bar{\mathcal{A}}}$ and $\boldsymbol{w}_{\bar{\mathcal{A}}} = \boldsymbol{0}_{\bar{\mathcal{A}}}$. Suppose that $\hat{\boldsymbol{w}}$ is not a critical point, the second equation is indeed converted to an inequality constraint via varying $\hat{\boldsymbol{t}}_{\bar{\mathcal{A}}}$ in $[-1, 1]^{|\bar{\mathcal{A}}|}$. As a result, (23) is merely in connection with $\boldsymbol{w}_{\mathcal{A}}$ and the left side in the first equation consists with the function $\mathcal{F}$ in Theorem 3. Take the *mixture* SP-regularizer as an example, the optimality of estimation when using mixture SP-regularizer is described as

$$\frac{1}{n}\sum_{i\in\mathcal{E}} (x_i \hat{\boldsymbol{w}}_{\mathcal{A}} - y_i) x_i^T + \frac{\gamma}{n} \sum_{k\in\mathcal{M}} \underbrace{\left(\frac{1}{2\sqrt{l_k}} - \frac{1}{\lambda}\right)(x_k \hat{\boldsymbol{w}}_{\mathcal{A}} - y_k) x_k^T}_{\mathcal{Z}(\hat{\boldsymbol{w}}_{\mathcal{A}},\lambda)} + \alpha \cdot \mathbf{sgn}(\hat{\boldsymbol{w}}_{\mathcal{A}}) = \boldsymbol{0}.$$

For the sake of simplicity, we derive the result of $\dfrac{d\mathcal{Z}(\hat{\boldsymbol{w}}_{\mathcal{A}}, \lambda)}{d\lambda}$ first. Note that $l_k = (x_k \hat{\boldsymbol{w}}_{\mathcal{A}} - y_k)^2$.

$$\frac{d\mathcal{Z}(\hat{\boldsymbol{w}}_{\mathcal{A}}, \lambda)}{d\lambda} = \left[-\frac{1}{4l_k\sqrt{l_k}} \cdot 2(x_k \hat{\boldsymbol{w}}_{\mathcal{A}} - y_k) \cdot x_k \frac{d\hat{\boldsymbol{w}}_{\mathcal{A}}}{d\lambda} + \frac{1}{\lambda^2}\right](x_k \hat{\boldsymbol{w}}_{\mathcal{A}} - y_k) x_k^T + \left(\frac{1}{2\sqrt{l_k}} - \frac{1}{\lambda}\right) x_k^T x_k \frac{d\hat{\boldsymbol{w}}_{\mathcal{A}}}{d\lambda}$$

$$= \frac{1}{\lambda^2}(x_k \hat{\boldsymbol{w}}_{\mathcal{A}} - y_k) x_k^T - \frac{1}{\lambda} x_k^T x_k \frac{d\hat{\boldsymbol{w}}_{\mathcal{A}}}{d\lambda}.$$

Similar to the proof of Theorem 3, we have

$$\frac{1}{n}\sum_{i\in\mathcal{E}} x_i^T x_i \frac{d\hat{\boldsymbol{w}}_{\mathcal{A}}}{d\lambda} + \frac{\gamma}{n}\sum_{k\in\mathcal{M}}\left[\frac{1}{\lambda^2}(x_k \hat{\boldsymbol{w}}_{\mathcal{A}} - y_k) x_k^T - \frac{1}{\lambda} x_k^T x_k \frac{d\hat{\boldsymbol{w}}_{\mathcal{A}}}{d\lambda}\right] = \boldsymbol{0}.$$

And final result comes from combining and vectoring the terms w.r.t. $\dfrac{d\hat{\boldsymbol{w}}_{\mathcal{A}}}{d\lambda}$, which can be utilized to derive the Prop. 2.

$\square$

## C  Detailed Algorithms

In this section, we present more details of the concrete algorithms derived for the SVM and Lasso.

### C.1  Support Vector Machines

The goal of GAGA is to calculate the age-path on the interval $[\lambda_{min}, \lambda_{max}]$. As mentioned in Section 3, started from an initial point, the algorithm solves the derived ODEs (8) while examining all the partitions along the way of $\lambda$. In SVM, partition violating is merely caused by the change in $\boldsymbol{g}$ (*c.f.* Section 4.1). For example, some $g_i$ varying from a negative value to zero will lead the violation of $\mathcal{E}_N$, resulting in a critical point. In case that the point belongs to a turning point, the only need is to resign $i$ from $\mathcal{E}_N$ to $\mathcal{E}_Z$. Otherwise the point is a jump point and we have to perform the warm start to re-calculate the next solution, which could be time consuming. Since it's non-trivial to identify the type of the critical point as a prior, we adopt a heuristic operation to avoid excessive warm starts. When the partition violation occurs, we directly resign all the violated indexes into the right status by the partition rule. Suppose that a turning point is encountered, the solutions return by numerical ODEs solver with the updated index sets will keep the KKT condition, allowing our algorithm to proceed. The algorithmic steps are given in Algorithm 3.

---

**Algorithm 3**  GAGA for SVM

---

**Input**: Initial solution $(\boldsymbol{\alpha}, b)|_{\lambda_t = \lambda_{min}}, X, y, \lambda_{min}$ and $\lambda_{max}$
**Parameter**: Cost parameter $C$
**Output**: Age-path $(\boldsymbol{\alpha}, b)$ on $[\lambda_{min}, \lambda_{max}]$

1: $\lambda_t \leftarrow \lambda_{min}$, set $\mathcal{E}_N, \mathcal{E}_Z, \mathcal{E}_P, \mathcal{D}(, \mathcal{M})$ in Proposition 1 according to $\boldsymbol{w}|_{\lambda_t = \lambda_{min}}$
2: **while** $\lambda_t \leq \lambda_{max}$ **do**
3:     Solve (8) or (9) and partition samples in $X$ and components of $\boldsymbol{\alpha}$ simultaneously.
4:     **if** Partition $\mathcal{E}_N, \mathcal{E}_Z, \mathcal{E}_P, \mathcal{D}(, \mathcal{M})$ was not met **then**
5:         Resign violated indexes in $\mathcal{P}$ by $g_i$.
6:     **end if**
7:     $\boldsymbol{\alpha}_{\mathcal{E}_N} = \boldsymbol{0}_{\mathcal{E}_N}, \boldsymbol{\alpha}_{\mathcal{D}} = \boldsymbol{0}_{\mathcal{D}}$. For the mixture regularizer, $\boldsymbol{\alpha}_{\mathcal{E}_P} = \boldsymbol{1}_{\mathcal{E}_P}$.
8:     Solve (8) or (9) with updated $\mathcal{E}_N, \mathcal{E}_Z, \mathcal{E}_P, \mathcal{D}(, \mathcal{M})$
9:     **if** KKT conditions are not met **then**
10:         Warm start at $\lambda_t + \delta$ (for a small $\delta > 0$).
11:     **end if**
12: **end while**

---

### C.2  Lasso

In Lasso, the main routine of GAGA is similar with that in SVM. The only difference is that here we need to examine the partition $\mathcal{A}$ additionally. Due to the property of the $\ell_1$ norm, monitoring $\mathcal{A}$ is operated by observing if the value of $\boldsymbol{w}_i$ equals to zero or conversely, whether the subgradient of inactive component is reached to 1. The details of the algorithms are shown in Algorithm 4.

## D  Additional Results

In this section, we present additional experimental results on the logistic regression and path consistency to obtain a more comprehensive evaluation of proposed algorithm.

### D.1  Logistic Regression

Given the dataset $X$ and label $\boldsymbol{y}$, the logistic regression gives the optimization problem as (24)

$$\min_{\boldsymbol{w} \in \mathbb{R}^d, \, b} \quad \frac{1}{2}\|\boldsymbol{w}\|^2 + \sum_{i=1}^{n} C \ln\left(1 + e^{-y_i(X_i \boldsymbol{w} + b)}\right), \tag{24}$$

where $C > 0$ is the trade-off parameter. Note the objective (24) is smooth on the entire domain, hence we apply Corollary 1 to derive the ODEs and the only $\mathcal{P}$ is needde to be tracked and reset

**Algorithm 4** `GAGA` for Lasso

---

**Input**: $\boldsymbol{w}|_{\lambda_t = \lambda_{min}}$, $X$, $y$, $\lambda_{min}$ and $\lambda_{max}$
**Parameter**: Regularization strength $\alpha$
**Output**: Age-path $\boldsymbol{w}$ on $[\lambda_{min}, \lambda_{max}]$

1: $\lambda_t \leftarrow \lambda_{min}$, set $\mathcal{A}, \mathcal{P}$ in Proposition 2 according to $\boldsymbol{w}|_{\lambda_t = \lambda_{min}}$
2: **while** $\lambda_t \leq \lambda_{max}$ **do**
3:     Solve (11) or (12) and partition $\boldsymbol{w}_{\mathcal{A}}$, $X_{\mathcal{A}}$ simultaneously.
4:     **if** Partition $\mathcal{A}, \mathcal{P}$ was not met **then**
5:         **if** $\mathcal{A}$ was not met **then**
6:             **if** $k$-th element turns to inactive **then**
7:                 $\boldsymbol{w}_k = \mathbf{0}$.
8:                 Remove $k$ from $\mathcal{A}$.
9:             **else if** $k$-th element becomes active **then**
10:                 Put $k$ into $\mathcal{A}$.
11:             **end if**
12:         **end if**
13:         **if** $\mathcal{P}$ was not met **then**
14:             Resign violated indexes in $\mathcal{P}$ by $\ell_i$.
15:         **end if**
16:         Solve (11) or (12) with updated $\mathcal{A}, \mathcal{P}$.
17:         **if** KKT conditions are not met **then**
18:             Warm start at $\lambda_t + \delta$ (for a small $\delta > 0$).
19:         **end if**
20:     **end if**
21: **end while**

---

along the path. To start with, let $\ell_i = C \ln(1 + e^{-y_i(X_i \boldsymbol{w} + b)})$, then for the linear SP-regularizer, the partition $\mathcal{P} = \{\mathcal{E}, \mathcal{D}\}$, where $\mathcal{E} = \{1 \leq i \leq n : \ell_i < \lambda\}$, $\mathcal{D} = \{1 \leq i \leq n : \ell_i \geq \lambda\}$. For the mixture SP-regularizer, $\mathcal{P} = \{\mathcal{E}, \mathcal{M}, \mathcal{D}\}$, where $\mathcal{E} = \{1 \leq i \leq n : \ell_i < \left(\frac{\lambda \gamma}{\lambda + \gamma}\right)^2\}$, $\mathcal{M} = \{1 \leq i \leq n : \left(\frac{\lambda \gamma}{\lambda + \gamma}\right)^2 \leq \ell_i \leq \lambda^2\}$, and $\mathcal{D} = \{1 \leq i \leq n : \ell_i > \lambda^2\}$. Applying Corollary 1 obtains Theorem 5.

**Theorem 5.** *When $\boldsymbol{w}, b$ indicate a partial optimum, the dynamics of optimal $\boldsymbol{w}, b$ in (24) w.r.t. $\lambda$ for the linear and mixture SP-regularizer are shown as*

$$
\frac{d\left(\begin{array}{c} \boldsymbol{w} \\ b \end{array}\right)}{d\lambda} = \left(\begin{array}{cc} I + C X_{\mathcal{E}}^T U_{\mathcal{E}} X_{\mathcal{E}} & C X_{\mathcal{E}}^T U_{\mathcal{E}} \\ C \mathbf{1}_{\mathcal{E}}^T U_{\mathcal{E}} X_{\mathcal{E}} & C \mathbf{1}_{\mathcal{E}}^T U_{\mathcal{E}} \end{array}\right)^{-1} \left(\begin{array}{c} C X_{\mathcal{E}}^T \left[\frac{\boldsymbol{y}_{\mathcal{E}} \odot \boldsymbol{\ell}_{\mathcal{E}}}{\lambda^2} \odot \left(e^{-\frac{\boldsymbol{\ell}_{\mathcal{E}}}{C}} - 1\right)\right] \\ C \mathbf{1}_{\mathcal{E}}^T \left[\frac{\boldsymbol{y}_{\mathcal{E}} \odot \boldsymbol{\ell}_{\mathcal{E}}}{\lambda^2} \odot \left(e^{-\frac{\boldsymbol{\ell}_{\mathcal{E}}}{C}} - 1\right)\right] \end{array}\right),
$$
$$(25)$$

*where* $U_{\mathcal{E}} = Diag\left\{\boldsymbol{y}_{\mathcal{E}}^2 \odot \boldsymbol{u}_{\mathcal{E}}\right\}$, $\boldsymbol{u}_{\mathcal{E}} = \left(\frac{\boldsymbol{\ell}_{\mathcal{E}} - C}{\lambda} - 1\right) \odot e^{-\frac{2\boldsymbol{\ell}_{\mathcal{E}}}{C}} + \left(\frac{2C - \boldsymbol{\ell}_{\mathcal{E}}}{\lambda} + 1\right) \odot e^{-\frac{\boldsymbol{\ell}_{\mathcal{E}}}{C}} - \frac{C}{\lambda}$.

$$
\frac{d\left(\begin{array}{c} \boldsymbol{w} \\ b \end{array}\right)}{d\lambda} = \left(\begin{array}{cc} I + C X_{\mathcal{A}}^T U_{\mathcal{A}} & C X_{\mathcal{A}} \\ C \mathbf{1}_{\mathcal{A}}^T U_{\mathcal{A}} X_{\mathcal{A}} & C \mathbf{1}_{\mathcal{A}}^T U_{\mathcal{A}} \end{array}\right)^{-1} \left(\begin{array}{c} C X_{\mathcal{A}}^T \left[\frac{\boldsymbol{y}_{\mathcal{A}} \odot \boldsymbol{\ell}_{\mathcal{A}}}{\lambda^2} \odot \left(e^{-\frac{\boldsymbol{\ell}_{\mathcal{A}}}{C}} - 1\right)\right] \\ C \mathbf{1}_{\mathcal{A}}^T \left[\frac{\boldsymbol{y}_{\mathcal{A}} \odot \boldsymbol{\ell}_{\mathcal{A}}}{\lambda^2} \odot \left(e^{-\frac{\boldsymbol{\ell}_{\mathcal{A}}}{C}} - 1\right)\right] \end{array}\right), \quad (26)
$$

*where* $\mathcal{A} = \mathcal{E} \cup \mathcal{M}, U_{\mathcal{A}} = Diag\left\{\boldsymbol{y}_{\mathcal{A}}^2 \odot \boldsymbol{u}_{\mathcal{A}}\right\}, \boldsymbol{u}_{\mathcal{E}} = e^{-\frac{\boldsymbol{\ell}_{\mathcal{E}}}{C}} \odot \left(1 - e^{-\frac{\boldsymbol{\ell}_{\mathcal{E}}}{C}}\right), \boldsymbol{u}_{\mathcal{M}} = \left(\frac{C}{2} \boldsymbol{\ell}_{\mathcal{M}}^{-\frac{3}{2}} + \boldsymbol{\ell}_{\mathcal{M}}^{-\frac{1}{2}} - \frac{1}{\lambda}\right) \odot e^{-\frac{2\boldsymbol{\ell}_{\mathcal{M}}}{C}} - \left(C \boldsymbol{\ell}_{\mathcal{M}}^{-\frac{3}{2}} + \boldsymbol{\ell}_{\mathcal{M}}^{-\frac{1}{2}} - \frac{1}{\lambda}\right) \odot e^{-\frac{\boldsymbol{\ell}_{\mathcal{M}}}{C}} + \frac{C}{2} \boldsymbol{\ell}_{\mathcal{M}}^{-\frac{3}{2}}$.

*Proof.* The proof is nearly the same as that of SVM and Lasso, hence we merely present the main structure in the following. The linear SP-regularizer is utilized during the derivation, while the proof of mixture SP-regularizer is quite similar.

Given a partial optimum $(\boldsymbol{w}, b)$ at $\lambda$, (5) is rewritten in detail with the form of

$$\boldsymbol{w} + C \sum_{i \in \mathcal{E}} \left( \frac{1}{1 + e^{-y_i(\boldsymbol{w}^T X_i + b)}} - 1 \right) X_i^T = \mathbf{0}$$

$$C \sum_{i \in \mathcal{E}} \left( \frac{1}{1 + e^{-y_i(\boldsymbol{w}^T X_i + b)}} - 1 \right) = 0.$$

Similarly, we set the $\mathcal{F}$ as

$$\mathcal{F} = \begin{pmatrix} \boldsymbol{w} + C \sum_{i \in \mathcal{E}} \left( \frac{1}{1 + e^{-y_i(\boldsymbol{w}^T X_i + b)}} - 1 \right) X_i^T \\ C \sum_{i \in \mathcal{E}} \left( \frac{1}{1 + e^{-y_i(\boldsymbol{w}^T X_i + b)}} - 1 \right) \end{pmatrix}$$

$$= \begin{pmatrix} \boldsymbol{w} + C X_{\mathcal{E}}^T \left( \boldsymbol{y}_{\mathcal{E}} \odot \left( 1 - \frac{\boldsymbol{\ell_{\mathcal{E}}}}{\lambda} \right) \odot \left( e^{-\frac{\boldsymbol{\ell_{\mathcal{E}}}}{C}} - 1 \right) \right) \\ C \mathbf{1}_{\mathcal{E}}^T \left( \boldsymbol{y}_{\mathcal{E}} \odot \left( 1 - \frac{\boldsymbol{\ell_{\mathcal{E}}}}{\lambda} \right) \odot \left( e^{-\frac{\boldsymbol{\ell_{\mathcal{E}}}}{C}} - 1 \right) \right) \end{pmatrix}.$$

Afterwards, the corresponding jaconbian is derived as

$$\boldsymbol{J}_{\mathcal{F},(\boldsymbol{w},b)} = \begin{pmatrix} I + C X_{\mathcal{E}}^T U_{\mathcal{E}} X_{\mathcal{E}} & C X_{\mathcal{E}}^T U_{\mathcal{E}} \\ C \mathbf{1}_{\mathcal{E}}^T U_{\mathcal{E}} X_{\mathcal{E}} & C \mathbf{1}_{\mathcal{E}}^T U_{\mathcal{E}} \end{pmatrix}$$

$$\boldsymbol{J}_{\mathcal{F},\lambda} = - \begin{pmatrix} C X_{\mathcal{E}}^T \left[ \frac{\boldsymbol{y}_{\mathcal{E}} \odot \boldsymbol{\ell_{\mathcal{E}}}}{\lambda^2} \odot \left( e^{-\frac{\boldsymbol{\ell_{\mathcal{E}}}}{C}} - 1 \right) \right] \\ C \mathbf{1}_{\mathcal{E}}^T \left[ \frac{\boldsymbol{y}_{\mathcal{E}} \odot \boldsymbol{\ell_{\mathcal{E}}}}{\lambda^2} \odot \left( e^{-\frac{\boldsymbol{\ell_{\mathcal{E}}}}{C}} - 1 \right) \right] \end{pmatrix},$$

where $U_{\mathcal{E}} = Diag\left\{ \boldsymbol{y}_{\mathcal{E}}^2 \odot \boldsymbol{u}_{\mathcal{E}} \right\}$, $\boldsymbol{u}_{\mathcal{E}} = \left( \frac{\boldsymbol{\ell_{\mathcal{E}}} - C}{\lambda} - 1 \right) \odot e^{-\frac{2\boldsymbol{\ell_{\mathcal{E}}}}{C}} + \left( \frac{2C - \boldsymbol{\ell_{\mathcal{E}}}}{\lambda} + 1 \right) \odot e^{-\frac{\boldsymbol{\ell_{\mathcal{E}}}}{C}} - \frac{C}{\lambda}$.
Therefore, the implicit function theorem implies that

$$\frac{d \begin{pmatrix} \boldsymbol{w} \\ b \end{pmatrix}}{d\lambda} = -\boldsymbol{J}_{\mathcal{F},(\boldsymbol{w},b)}^{-1} \cdot \boldsymbol{J}_{\mathcal{F},\lambda}$$

$$= \begin{pmatrix} I + C X_{\mathcal{E}}^T U_{\mathcal{E}} X_{\mathcal{E}} & C X_{\mathcal{E}}^T U_{\mathcal{E}} \\ C \mathbf{1}_{\mathcal{E}}^T U_{\mathcal{E}} X_{\mathcal{E}} & C \mathbf{1}_{\mathcal{E}}^T U_{\mathcal{E}} \end{pmatrix}^{-1} \begin{pmatrix} C X_{\mathcal{E}}^T \left[ \frac{\boldsymbol{y}_{\mathcal{E}} \odot \boldsymbol{\ell_{\mathcal{E}}}}{\lambda^2} \odot \left( e^{-\frac{\boldsymbol{\ell_{\mathcal{E}}}}{C}} - 1 \right) \right] \\ C \mathbf{1}_{\mathcal{E}}^T \left[ \frac{\boldsymbol{y}_{\mathcal{E}} \odot \boldsymbol{\ell_{\mathcal{E}}}}{\lambda^2} \odot \left( e^{-\frac{\boldsymbol{\ell_{\mathcal{E}}}}{C}} - 1 \right) \right] \end{pmatrix}.$$

$\square$

## D.2 Detailed Experimental Setting

We use the Scikit-learn package [57] to optimize the subproblems of SVM, logistic regression and Lasso. The MOSPL method is implemented using the toolbox geatpy [58]. All codes were implemented in Python and all experiments were conducted on a machine with 48 2.2GHz cores, 80GB of RAM and 4 Nvidia 1080ti GPUs.

In all experiments of performance comparison, we evaluate the average performance in 20 runs. To maintain the reproducibility, the random seed is fixed with 40. In each trail, the each dataset is randomly divided into a training set and a testing set by the ratio of $3:1$. When carrying out `GAGA` and ACS, the predefined interval of $\lambda$ is set to $[0.1, 20]$, and the step size in ACS equals to $0.5$. We

utilize the NSGA-III as the framework of MOSPL, in which $N_p$ is set to 150 and $Gen = 800$[11]. When applying the mixture regularizer, we utilize the polynomials loss in [19] as $\ell_i$ to transform the original problem into a multi-objective problem. Afterwards, the polynomial order $t$ is fixed at 1.2 and 1.35, respectively.

### D.3    Simulation Study on Logistic Regression

We present additional experimental results on the logistic regression (24) to validate our ODEs. The utilized datasets are listed in Table 4. The averaged results using the linear and mixture SP-

| Dataset | Source | Samples | Dimensions | Task |
|---|---|---|---|---|
| mfeat-pixel | UCI | 2000 | 240 | |
| pendigits | UCI | 3498 | 16 | |
| hiva agnostic | OpenML | 4230 | 1620 | C |
| nomao | OpenML | 34465 | 118 | |
| MagicTelescope | OpenML | 19020 | 11 | |

Table 4: Datasets description in experiments on logistic regression. The C = Classification.

regularizer are illustrated in Table 5 and 6, respectively, in which the performance is measured by the classification accuracy. Meanwhile, Figure 7 confirms the computational efficiency of `GAGA` on large-scale dataset. Taking all results into consideration, `GAGA` outperforms than the baseline methods on all datasets and parameter settings, hence demonstrates the performance of `GAGA` in classification tasks with large data size.

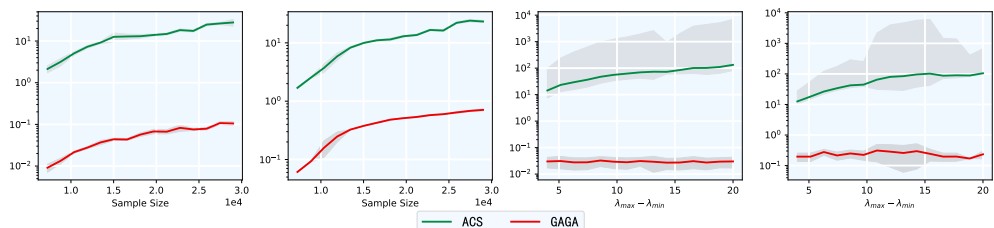

Figure 7: The study of efficiency comparison. $y$-axis denotes the average running time (in seconds) in 20 runs. The interval $[\lambda_{min}, \lambda_{max}]$ refers to the predefined search space.

### D.4    Path consistency

In this section we illustrate that the age-path tracked by `GAGA` exactly consists with the path of real partial optimum, which is produced by the ACS algorithm. Due to the expensive computational cost of finding the partial optimum using ACS (over 30 loops on average), we choose the toy datasets from the Scikit-learn package to trace and plot the age-path. In detail, we use the Boston house and breast cancer datasets for regression tasks, and the classification is performed on the handwritten digits dataset. In order to track the exact path of partial optimum, we set the step size to be 1e-4 and 3e-1 in ACS and `GAGA`, respectively. The graphs of the tracked path are illustrated in Figure 8, Figure 9 and Figure 10. The path of partial optimum is plotted in blue solid lines while the age-path traced by `GAGA` is marked with red dashed lines.

This result empirically validates the path consistency between the computed age-path by `GAGA` and the ground truth age-path (*i.e.* path of the partial optimum), which is stated in Theorem (3).

### D.5    More Histograms

We further show the intrinsic property of the age-path in more experiments. Specifically, we change the dataset, SP-regularizer as well as the value of other hyper-parameter in SVM, Lasso and logistic regression, while recording the number of different types of critical points. The results are illustrated in Figure 11, 12 and 13.

---

[11]$N_p$ and $Gen$ represent the number of populations and the utmost generations in evolutionary algorithm.

Table 5: Average results with the standard deviation in 20 runs on different datasets using the *linear SP-regularizer*. The top results in each row are in boldface. The † and ‡ share the same meaning as in the main body.

| Dataset | Parameter | | Competing Methods | | | Ours | Restarting times |
|---------|-----------|--|-------------------|--|--|------|------------------|
| | $C$ | $\gamma$ | Original | ACS | MOSPL | GAGA | |
| mfeat-pixel† | 0.50 | – | 0.827±0.028 | 0.937±0.054 | 0.962±0.0281 | **0.980**±0.015 | 189 |
| pendigits‡ | 0.50 | – | 0.982±0.007 | 0.989±0.006 | 0.986±0.012 | **0.992**±0.006 | 86 |
| hiva agnostic† | 1.00 | – | 0.681±0.020 | 0.700±0.021 | 0.958±0.010 | **0.965**±0.004 | 328 |
| MagicTelescope† | 0.50 | – | 0.974±0.004 | 0.981±0.001 | 0.977±0.005 | **0.991**±0.001 | 61 |
| nomao‡ | 0.50 | – | 0.939±0.002 | 0.940±0.003 | 0.944±0.001 | **0.944**±0.001 | 32 |

Table 6: Average results with the standard deviation in 20 runs on different datasets using the *mixture SP-regularizer*. The top results in each row are in boldface. The † and ‡ share the same meaning as in the main body.

| Dataset | Parameter | | Competing Methods | | | Ours | Restarting times |
|---------|-----------|--|-------------------|--|--|------|------------------|
| | $C$ | $\gamma$ | Original | ACS | MOSPL | GAGA | |
| mfeat-pixel† | 0.50 | 0.20 | 0.827±0.028 | 0.980±0.011 | 0.981±0.014 | **0.981**±0.018 | 166 |
| pendigits‡ | 0.50 | 0.20 | 0.982±0.007 | 0.989±0.007 | 0.988±0.006 | **0.993**±0.005 | 178 |
| hiva agnostic† | 0.50 | 0.20 | 0.681±0.020 | 0.713±0.021 | 0.944±0.008 | **0.973**±0.014 | 195 |
| MagicTelescope† | 0.50 | 0.20 | 0.974±0.004 | 0.976±0.003 | 0.991±0.003 | **0.991**±0.001 | 73 |
| nomao‡ | 0.50 | 0.20 | 0.939±0.002 | 0.941±0.001 | 0.941±0.002 | **0.946**±0.002 | 44 |

These histograms demonstrate that there exists more turning points on the age-path compared with the jump points. As a result, the heuristic technique applied in GAGA can avoid extensive unnecessary warm starts by identifying the exact type of each critical point.

### D.6 Experimental Comparison of robust SVMs and Lasso

We also conduct a comparative study to the state-of-the-art robust model for SVMs [48, 49, 50] and Lasso [51] besides the SPL domain. Please take notice that RLSSVM and Re-LSSVM proposed by [48, 49] are variations of LS-SVM. Especially, we implement the Huber-loss Lasso as a special case of the generalized model proposed in [51]. The hyperparameters of these baselines are chosen from the best performance by grid search. All the experiments conducted on regression tasks are measured by the generalization error. We'd like to emphasize again that our GAGA framework pursues the best practice of conventional SPL while the SPL, as a special case of curriculum learning naturally owns certain shortcomings on the sample diversity [13]. Even subject to the inherent defects of vanilla SPL, the result in Table.7 reveals that our method still surpassed SOTA baselines in half of the experiments. The Table.8 further implies that GAGA outperforms the robust SVM in all conducted trails. These numerical results strongly validate the performance of our method under various noise levels.

Concretely speaking, the linear SP-regularizer is utilized in GAGA in the implementation. The $\gamma_1$ and $\gamma_2$ are hyperparameters of **RLSSVM** and **RE-LSSVM**. We use the same hyperparameters of GAGA in Appendix D.7.

### D.7 Sensitivity Analysis

In this subsection, we use the same settings except the backbone parameters $\alpha$, $C$ and the noise level. The linear SP-regularizer is utilized in GAGA. The classical SVM with linear kernel is chosen as the base model. Results in Table 9, 10, 11, 12 confirm the performances of competing methods with different backbone parameters while retaining the same noise level. Table 13, 14, 15, 16 display the running results under different noise levels while the other backbone parameters are kept. The results of the massive simulation studies again strongly demonstrate that our GAGA owns the best practice of the conventional SPL compared with the baseline methods, regardless of the specific parametric selections.

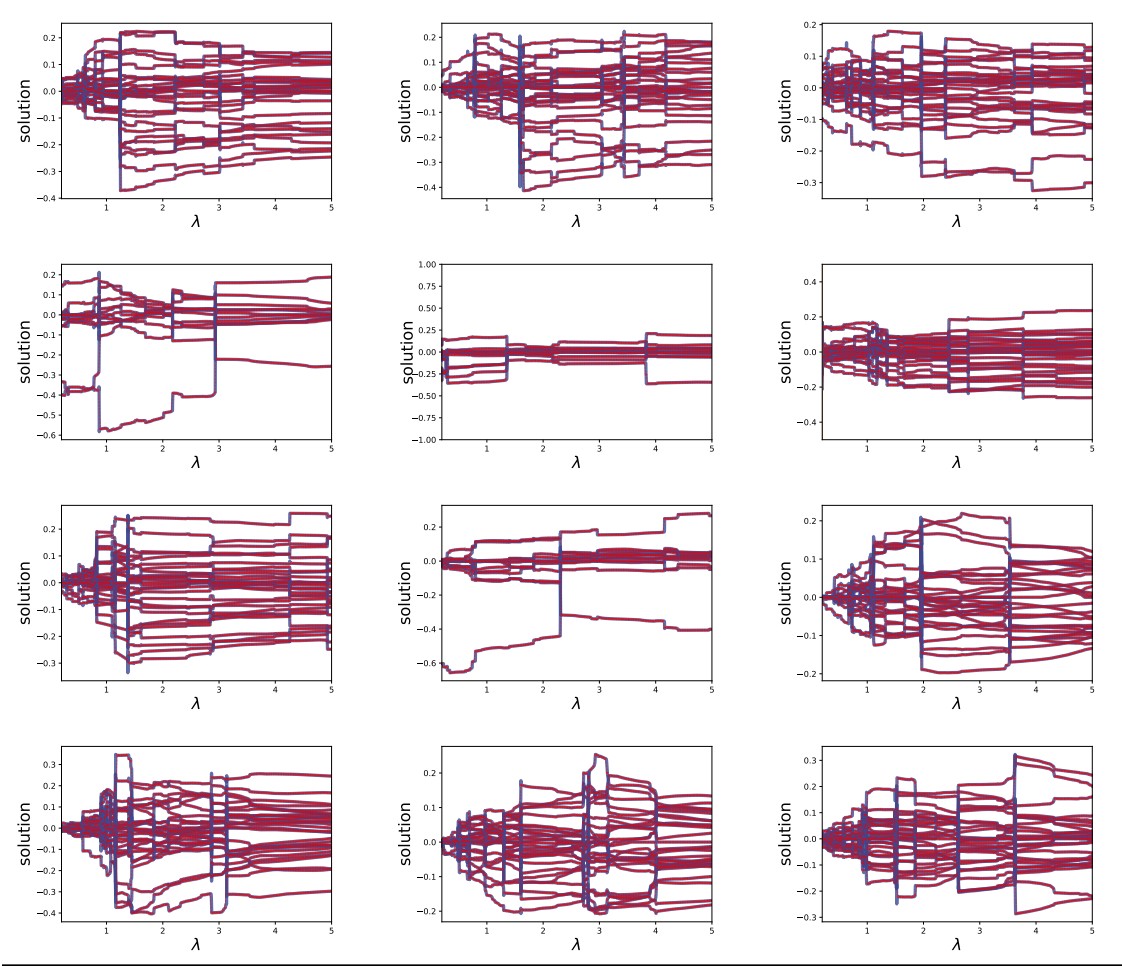

Figure 8: Age-path of SVM with different parameters and datasets. The first two rows of subfigures illustrate age-path using the linear SP-regularizer while the last two rows of subfigures show age-path using the mixture SP-regularizer. For experiments in the first and third row, the $\alpha = 0.02$. For experiments in the second and fourth row, the $\alpha = 0.04$.

| Parameter | | | | | | Huber Lasso | | RLSSVM | | Re-LSSVM | | GAGA | | Noise | Dataset |
|---|---|---|---|---|---|---|---|---|---|---|---|---|---|---|---|
| $e$ | $\alpha$ | $\sigma$ | $\gamma_1$ | $\theta$ | $\gamma_2$ | Mean | Std | Mean | Std | Mean | Std | Mean | Std | | |
| 0.2 | 0.0006 | 0.7 | 1.9 | 0.2 | 1.9 | 0.399 | 0.003 | 0.449 | 0.107 | **0.318** | 0.010 | 0.491 | 0.001 | 0.1 | ailerons |
| 0.1 | 0.0001 | 0.8 | 1.9 | 0.4 | 1.8 | **0.410** | 0.003 | 0.573 | 0.082 | 0.499 | 0.010 | 0.49 | 0.002 | 0.2 | ailerons |
| 0.2 | 0.0016 | 0.5 | 1.9 | 0.2 | 1.9 | **0.407** | 0.003 | 0.419 | 0.079 | 0.696 | 0.018 | 0.489 | 0.002 | 0.3 | ailerons |
| 0.2 | 0.0006 | 0.7 | 1.9 | 0.9 | 1.8 | **0.428** | 0.005 | 0.500 | 0.050 | 0.877 | 0.014 | 0.491 | 0.001 | 0.4 | ailerons |
| 0.1 | 0.0008 | 0.8 | 1.9 | 0.2 | 1.9 | **0.452** | 0.003 | 0.705 | 0.052 | 1.118 | 0.017 | 0.491 | 0.001 | 0.5 | ailerons |
| 0.2 | 0.0018 | 0.8 | 1.9 | 0.3 | 1.7 | **0.478** | 0.003 | 0.565 | 0.044 | 1.159 | 0.025 | 0.490 | 0.015 | 0.6 | ailerons |
| 0.1 | 0.0002 | 0.8 | 1.9 | 0.2 | 1.7 | **0.180** | 0.000 | 0.207 | 0.082 | 0.315 | 0.024 | 0.201 | 0.046 | 0.2 | houses |
| 0.1 | 0.0013 | 0.5 | 1.9 | 0.9 | 1.8 | **0.182** | 0.000 | 0.206 | 0.042 | 0.343 | 0.170 | 0.221 | 0.015 | 0.3 | houses |
| 0.2 | 0.001 | 0.8 | 1.7 | 0.8 | 1.9 | 0.345 | 0.002 | 0.303 | 0.083 | 0.387 | 0.014 | **0.214** | 0.002 | 0.1 | music |
| 0.1 | 0.0013 | 0.5 | 1.9 | 0.6 | 1.9 | 0.365 | 0.001 | 0.315 | 0.011 | 0.486 | 0.013 | **0.213** | 0.005 | 0.2 | music |
| 0.2 | 0.0011 | 0.9 | 1.9 | 0.4 | 1.8 | 0.357 | 0.002 | 0.308 | 0.057 | 0.444 | 0.057 | **0.214** | 0.003 | 0.3 | music |
| 0.1 | 0.0019 | 1 | 1.9 | 0.4 | 1.7 | 0.406 | 0.003 | 0.422 | 0.087 | 0.412 | 0.015 | **0.211** | 0.008 | 0.4 | music |
| 0.2 | 0.0017 | 0.5 | 1.9 | 0.6 | 1.8 | 0.431 | 0.004 | 0.489 | 0.015 | 0.477 | 0.035 | **0.210** | 0.003 | 0.5 | music |
| 0.3 | 0.0011 | 0.8 | 1.9 | 0.3 | 1.9 | 0.484 | 0.005 | 0.598 | 0.017 | 0.521 | 0.019 | **0.214** | 0.001 | 0.6 | music |
| 0.1 | 0.0005 | 0.7 | 1.9 | 0.8 | 1.9 | 0.663 | 0.000 | 0.747 | 0.089 | 0.689 | 0.010 | **0.647** | 0.132 | 0.2 | delta elevators |
| 0.1 | 0.0014 | 0.8 | 1.9 | 0.4 | 1.8 | 0.666 | 0.000 | 0.744 | 0.025 | 0.694 | 0.033 | **0.634** | 0.132 | 0.3 | delta elevators |

Table 7: Average generalization erros with the standard deviation in 20 runs on different datasets. The top results in each row are in boldface.

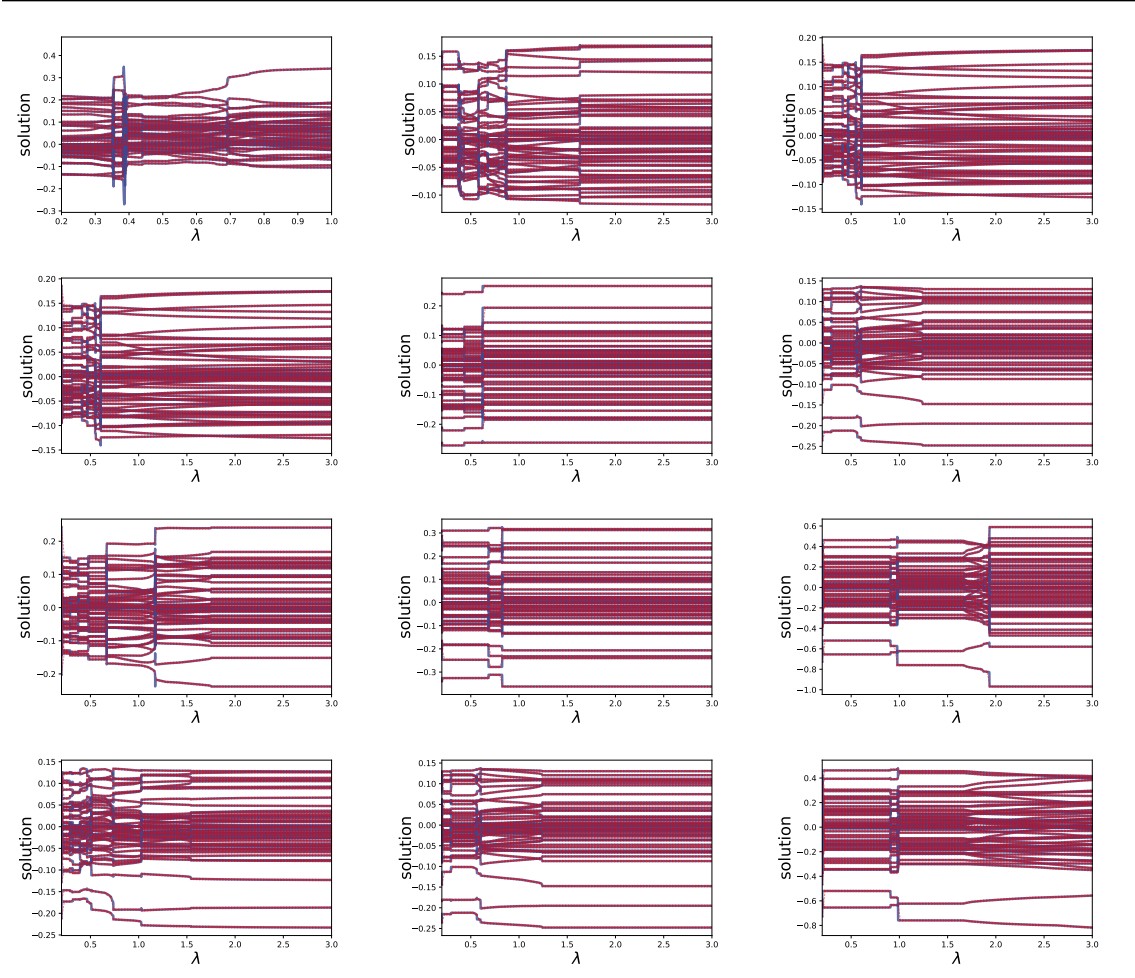

Figure 9: Age-path of Lasso with different parameters and datasets. The first two rows of subfigures illustrate age-path using the linear SP-regularizer while the last two rows of subfigures show age-path using the mixture SP-regularizer. For experiments in the first and third row, the $C = 0.02$. For experiments in the second and fourth row, the $C = 0.04$.

| Dataset | Parameter | | Robust-SVM | | GAGA | | Noise Level |
|---|---|---|---|---|---|---|---|
| | C | $\gamma$ | Mean | Std | Mean | Std | |
| mfeat-pixel | 1 | 0.8 | 0.948 | 0.001 | **0.980** | **0.006** | 0.1 |
| mfeat-pixel | 1 | 0.2 | 0.939 | 0.001 | **0.988** | **0.007** | 0.2 |
| mfeat-pixel | 1 | 0.7 | 0.927 | 0.002 | **0.980** | **0.015** | 0.3 |
| pendigts | 1 | 0.7 | 0.945 | 0.003 | **0.998** | **0.007** | 0.1 |
| pendigts | 1 | 0.6 | 0.944 | 0.002 | **0.995** | **0.004** | 0.2 |
| pendigts | 1 | 0.4 | 0.923 | 0.001 | **0.995** | **0.006** | 0.3 |

Table 8: Average classification accuracy with the standard deviation in 20 runs under different noise levels. The top results in each row are in boldface.

| Dataset | Parameter $\alpha$ | Competing Methods ACS | | MOSPL | | GAGA | | Noise Level |
|---|---|---|---|---|---|---|---|---|
| ailerons | 0.001 | 0.494 | 0.001 | 0.493 | 0.008 | **0.490** | **0.001** | 0.3 |
| ailerons | 0.002 | 0.493 | 0.007 | 0.492 | 0.016 | **0.490** | **0.002** | 0.3 |
| ailerons | 0.003 | 0.493 | 0.001 | 0.493 | 0.001 | **0.489** | **0.002** | 0.3 |
| ailerons | 0.004 | 0.493 | 0.001 | 0.492 | 0.001 | **0.491** | **0.001** | 0.3 |
| ailerons | 0.005 | 0.493 | 0.001 | 0.493 | 0.001 | **0.49** | **0.001** | 0.3 |
| ailerons | 0.006 | 0.493 | 0.001 | 0.493 | 0.001 | **0.491** | **0.001** | 0.3 |
| ailerons | 0.007 | 0.494 | 0.002 | 0.492 | 0.002 | **0.491** | **0.001** | 0.3 |
| ailerons | 0.008 | 0.493 | 0.002 | 0.493 | 0.002 | **0.493** | **0.002** | 0.3 |
| ailerons | 0.009 | 0.493 | 0.002 | 0.493 | 0.002 | **0.493** | **0.002** | 0.3 |
| ailerons | 0.010 | 0.493 | 0.002 | 0.493 | 0.002 | **0.493** | **0.002** | 0.3 |
| ailerons | 0.011 | 0.493 | 0.002 | 0.493 | 0.002 | **0.493** | **0.002** | 0.3 |
| ailerons | 0.012 | 0.493 | 0.002 | 0.493 | 0.002 | **0.493** | **0.002** | 0.3 |
| ailerons | 0.013 | 0.493 | 0.002 | 0.493 | 0.002 | **0.493** | **0.002** | 0.3 |
| ailerons | 0.014 | 0.493 | 0.002 | 0.493 | 0.002 | **0.493** | **0.002** | 0.3 |
| ailerons | 0.015 | 0.493 | 0.002 | 0.493 | 0.002 | **0.493** | **0.002** | 0.3 |

Table 9: Average generalization errors and the standard deviation in 20 runs with different values of $\alpha$. The top results in each row are in boldface.

| Dataset | Parameter $\alpha$ | Competing Methods ACS | | MOSPL | | GAGA | | Noise Level |
|---|---|---|---|---|---|---|---|---|
| music | 0.001 | 0.230 | 0.011 | **0.226** | 0.009 | 0.227 | **0.011** | 0.3 |
| music | 0.002 | 0.224 | 0.005 | 0.222 | 0.006 | **0.218** | **0.004** | 0.3 |
| music | 0.003 | 0.227 | 0.011 | 0.223 | 0.008 | **0.219** | **0.009** | 0.3 |
| music | 0.004 | 0.221 | 0.011 | 0.219 | 0.010 | **0.213** | **0.009** | 0.3 |
| music | 0.005 | 0.221 | 0.005 | 0.220 | 0.004 | **0.209** | **0.005** | 0.3 |
| music | 0.006 | 0.218 | 0.009 | 0.216 | 0.009 | **0.213** | **0.027** | 0.3 |
| music | 0.007 | 0.222 | 0.003 | 0.215 | 0.005 | **0.211** | **0.004** | 0.3 |
| music | 0.008 | 0.215 | 0.004 | 0.213 | 0.003 | **0.208** | **0.002** | 0.3 |
| music | 0.009 | 0.216 | 0.006 | 0.213 | 0.005 | **0.207** | **0.003** | 0.3 |
| music | 0.010 | 0.216 | 0.007 | 0.214 | 0.007 | **0.208** | **0.003** | 0.3 |
| music | 0.011 | 0.215 | 0.008 | 0.212 | 0.006 | **0.206** | **0.005** | 0.3 |
| music | 0.012 | 0.211 | 0.007 | 0.208 | 0.007 | **0.205** | **0.005** | 0.3 |
| music | 0.013 | 0.210 | 0.005 | 0.210 | 0.003 | **0.207** | **0.003** | 0.3 |
| music | 0.014 | 0.212 | 0.004 | 0.208 | 0.005 | **0.206** | **0.003** | 0.3 |
| music | 0.015 | 0.211 | 0.006 | 0.210 | 0.004 | **0.207** | **0.003** | 0.3 |
| music | 0.016 | 0.211 | 0.006 | 0.211 | 0.007 | **0.206** | **0.004** | 0.3 |
| music | 0.017 | 0.207 | 0.004 | 0.207 | 0.006 | **0.205** | **0.004** | 0.3 |
| music | 0.018 | 0.210 | 0.006 | 0.206 | 0.007 | **0.205** | **0.005** | 0.3 |
| music | 0.019 | 0.210 | 0.006 | 0.208 | 0.007 | **0.206** | **0.003** | 0.3 |
| music | 0.020 | 0.206 | 0.003 | 0.206 | 0.003 | **0.206** | **0.003** | 0.3 |
| music | 0.021 | 0.207 | 0.005 | 0.207 | 0.005 | **0.205** | **0.003** | 0.3 |
| music | 0.022 | 0.207 | 0.004 | 0.207 | 0.003 | **0.205** | **0.002** | 0.3 |
| music | 0.023 | 0.206 | 0.004 | 0.205 | 0.003 | **0.205** | **0.003** | 0.3 |
| music | 0.024 | 0.206 | 0.003 | 0.207 | 0.005 | **0.205** | **0.004** | 0.3 |
| music | 0.025 | 0.207 | 0.003 | 0.207 | 0.003 | **0.204** | **0.003** | 0.3 |
| music | 0.026 | 0.206 | 0.002 | 0.206 | 0.003 | **0.206** | **0.003** | 0.3 |
| music | 0.027 | 0.208 | 0.004 | 0.206 | 0.004 | **0.205** | **0.004** | 0.3 |
| music | 0.028 | 0.208 | 0.005 | 0.208 | 0.005 | **0.205** | **0.004** | 0.3 |
| music | 0.029 | 0.205 | 0.005 | 0.205 | 0.005 | **0.205** | **0.003** | 0.3 |
| music | 0.03 | 0.207 | 0.004 | 0.207 | 0.004 | **0.205** | **0.002** | 0.3 |

Table 10: Average generalization errors and the standard deviation in 20 runs with different values of $\alpha$. The top results in each row are in boldface.

| Dataset | Parameter C | Competing Methods ACS | | MOSPL | | GAGA | | Noise Level |
|---|---|---|---|---|---|---|---|---|
| mfeat-pixel | 0.100 | 0.932 | 0.012 | 0.959 | 0.005 | **0.965** | **0.005** | 0.3 |
| mfeat-pixel | 0.200 | 0.929 | 0.009 | 0.957 | 0.006 | **0.966** | **0.003** | 0.3 |
| mfeat-pixel | 0.300 | 0.937 | 0.054 | 0.962 | 0.281 | **0.980** | **0.015** | 0.3 |
| mfeat-pixel | 0.400 | 0.936 | 0.011 | 0.966 | 0.013 | **0.979** | **0.014** | 0.3 |
| mfeat-pixel | 0.500 | 0.928 | 0.008 | 0.951 | 0.004 | **0.961** | **0.005** | 0.3 |
| mfeat-pixel | 0.600 | 0.933 | 0.008 | 0.960 | 0.006 | **0.967** | **0.004** | 0.3 |
| mfeat-pixel | 0.700 | 0.922 | 0.012 | 0.946 | 0.006 | **0.948** | **0.005** | 0.3 |
| mfeat-pixel | 0.800 | 0.927 | 0.014 | 0.952 | 0.008 | **0.960** | **0.002** | 0.3 |
| mfeat-pixel | 0.900 | 0.927 | 0.006 | 0.945 | 0.010 | **0.955** | **0.010** | 0.3 |
| mfeat-pixel | 1.000 | 0.925 | 0.007 | 0.948 | 0.006 | **0.951** | **0.001** | 0.3 |
| mfeat-pixel | 1.100 | 0.928 | 0.009 | 0.952 | 0.006 | **0.955** | **0.002** | 0.3 |
| mfeat-pixel | 1.200 | 0.933 | 0.018 | 0.961 | 0.009 | **0.972** | **0.005** | 0.3 |
| mfeat-pixel | 1.300 | 0.929 | 0.011 | 0.953 | 0.009 | **0.964** | **0.004** | 0.3 |
| mfeat-pixel | 1.400 | 0.934 | 0.007 | 0.966 | 0.013 | **0.971** | **0.004** | 0.3 |
| mfeat-pixel | 1.500 | 0.936 | 0.007 | 0.958 | 0.013 | **0.978** | **0.003** | 0.3 |
| mfeat-pixel | 1.600 | 0.920 | 0.007 | 0.923 | 0.007 | **0.945** | **0.004** | 0.3 |
| mfeat-pixel | 1.700 | 0.931 | 0.010 | 0.948 | 0.014 | **0.969** | **0.004** | 0.3 |
| mfeat-pixel | 1.800 | 0.934 | 0.015 | 0.963 | 0.015 | **0.970** | **0.005** | 0.3 |
| mfeat-pixel | 1.900 | 0.929 | 0.007 | 0.957 | 0.006 | **0.962** | **0.007** | 0.3 |

Table 11: Average classification accuracy and the standard deviation in 20 runs with different values of C. The top results in each row are in boldface.

| Dataset | Parameter C | Competing Methods ACS | | MOSPL | | GAGA | | Noise Level |
|---|---|---|---|---|---|---|---|---|
| pendigts | 0.100 | 0.984 | 0.003 | 0.984 | 0.006 | **0.998** | **0.007** | 0.3 |
| pendigts | 0.200 | 0.989 | 0.003 | 0.991 | 0.008 | **0.995** | **0.004** | 0.3 |
| pendigts | 0.3 | 0.983 | 0.004 | 0.989 | 0.009 | **0.995** | **0.006** | 0.3 |
| pendigts | 0.400 | 0.985 | 0.002 | 0.988 | 0.007 | **0.992** | **0.006** | 0.3 |
| pendigts | 0.500 | 0.982 | 0.005 | 0.983 | 0.005 | **0.993** | **0.004** | 0.3 |
| pendigts | 0.600 | 0.987 | 0.007 | 0.989 | 0.010 | **0.992** | **0.004** | 0.3 |
| pendigts | 0.700 | 0.986 | 0.007 | 0.988 | 0.006 | **0.996** | **0.002** | 0.3 |
| pendigts | 0.800 | 0.979 | 0.003 | 0.987 | 0.009 | **0.995** | **0.007** | 0.3 |
| pendigts | 0.900 | 0.987 | 0.003 | 0.985 | 0.011 | **0.991** | **0.005** | 0.3 |
| pendigts | 1.000 | 0.989 | 0.006 | 0.986 | 0.012 | **0.992** | **0.006** | 0.3 |
| pendigts | 1.100 | 0.982 | 0.005 | 0.987 | 0.005 | **0.990** | **0.554** | 0.3 |
| pendigts | 1.200 | 0.983 | 0.008 | 0.982 | 0.007 | **0.994** | **0.004** | 0.3 |
| pendigts | 1.300 | 0.981 | 0.006 | 0.991 | 0.007 | **0.995** | **0.004** | 0.3 |
| pendigts | 1.400 | 0.980 | 0.003 | 0.983 | 0.002 | **0.989** | **0.005** | 0.3 |
| pendigts | 1.500 | 0.981 | 0.007 | 0.986 | 0.006 | **0.995** | **0.002** | 0.3 |
| pendigts | 1.600 | 0.981 | 0.009 | 0.985 | 0.011 | **0.992** | **0.003** | 0.3 |
| pendigts | 1.700 | 0.979 | 0.006 | 0.981 | 0.004 | **0.995** | **0.006** | 0.3 |
| pendigts | 1.800 | 0.977 | 0.004 | 0.983 | 0.005 | **0.993** | **0.006** | 0.3 |

Table 12: Average classification accuracy and the standard deviation in 20 runs with different values of C. The top results in each row are in boldface.

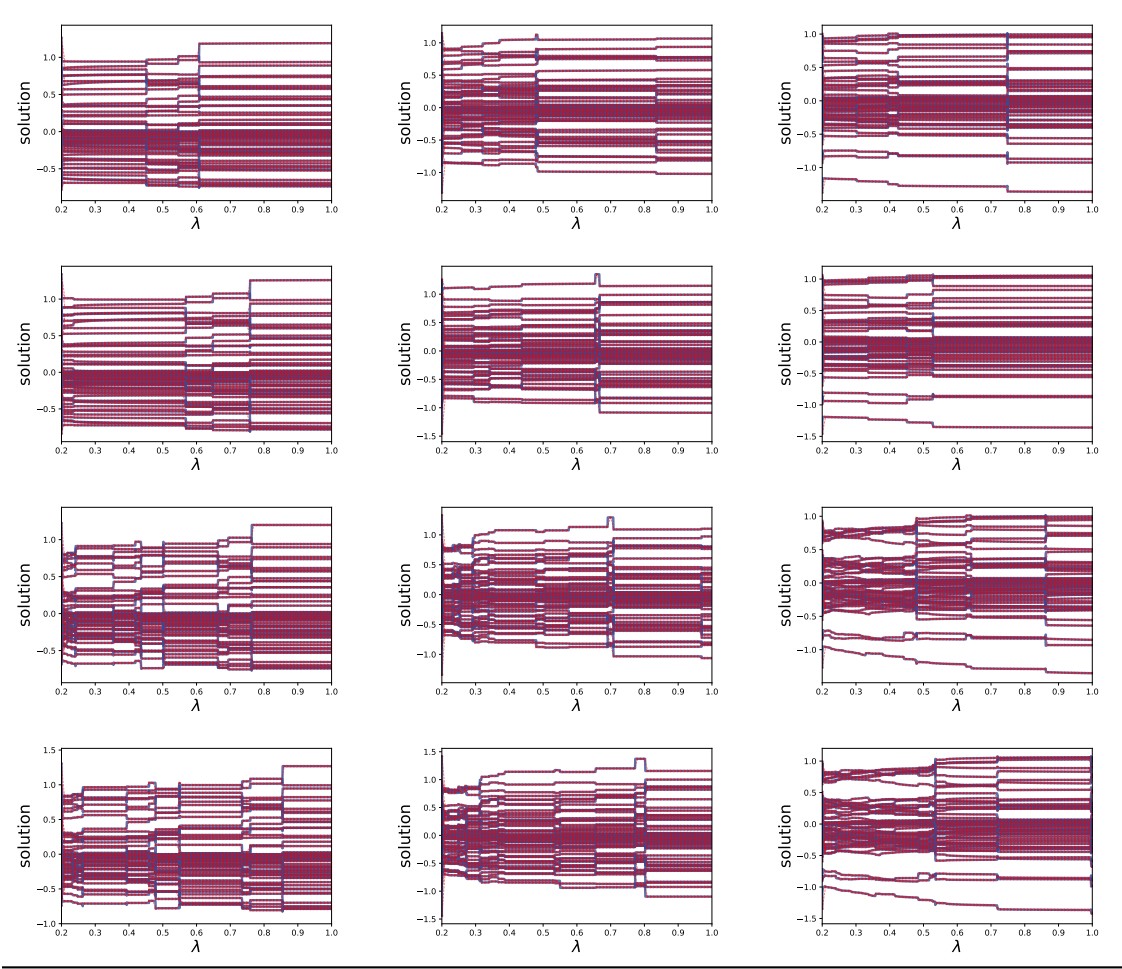

Figure 10: Age-path of logistic regression with different parameters and datasets. The first two rows of subfigures illustrate age-path using the linear SP-regularizer while the last two rows of subfigures show age-path using the mixture SP-regularizer. For experiments in the first and third row, the $C = 25$. For experiments in the second and fourth row, the $C = 40$.

| Dataset | Parameter | Competing Methods | | | | | | Noise Level |
|---|---|---|---|---|---|---|---|---|
| | $\alpha$ | ACS | | MOSPL | | GAGA | | |
| ailerons | 0.006 | 0.492 | 0.001 | 0.492 | 0.005 | **0.491** | **0.001** | 0.1 |
| ailerons | 0.006 | 0.492 | 0.001 | 0.491 | 0.016 | **0.49** | **0.002** | 0.2 |
| ailerons | 0.006 | 0.493 | 0.001 | 0.493 | 0.001 | **0.489** | **0.002** | 0.3 |
| ailerons | 0.006 | 0.493 | 0.001 | 0.492 | 0.001 | **0.491** | **0.001** | 0.4 |
| ailerons | 0.006 | 0.493 | 0.001 | 0.493 | 0.001 | **0.491** | **0.001** | 0.5 |
| ailerons | 0.006 | 0.493 | 0.001 | 0.492 | 0.001 | **0.491** | **0.001** | 0.6 |
| ailerons | 0.006 | 0.494 | 0.002 | 0.492 | 0.002 | **0.491** | **0.001** | 0.7 |
| ailerons | 0.006 | 0.493 | 0.002 | 0.493 | 0.002 | **0.493** | **0.002** | 0.8 |

Table 13: Average generalization errors with the standard deviation in 20 runs under different noise levels. The top results in each row are in boldface.

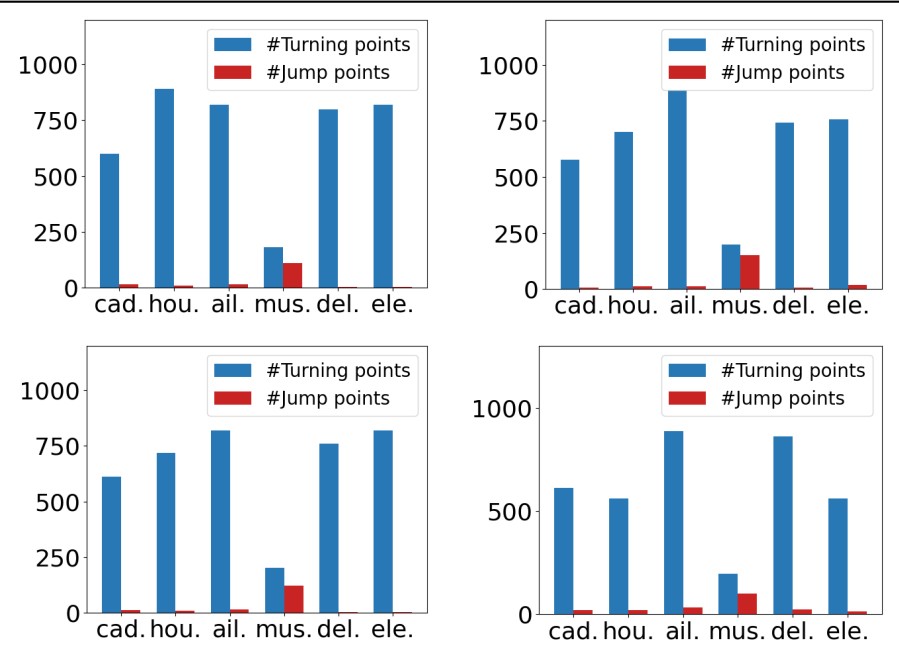

Figure 11: The number of different types of critical points in Lasso with different parameters and regularizers. The first row of subfigures illustrate different critical points in the age-path with linear SP-regularizer while the last row of figures show the number of different critical points in age-path using the mixture SP-regularizer.

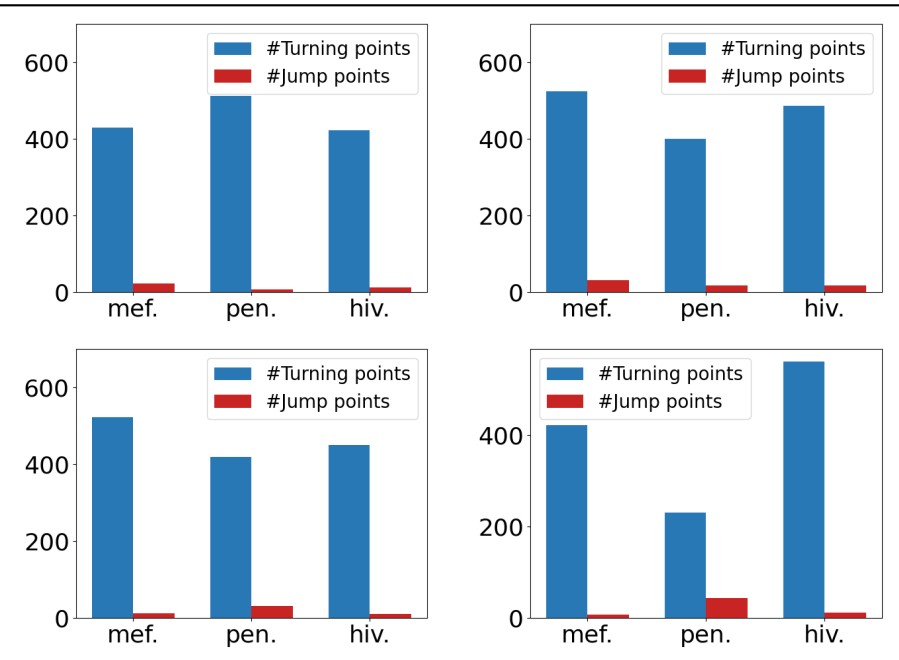

Figure 12: The number of different types of critical points in classic SVM with different parameters and regularizers.The first row of subfigures illustrate different critical points in the age-path with linear SP-regularizer while the last row of figures show the number of different critical points in age-path using the mixture SP-regularizer

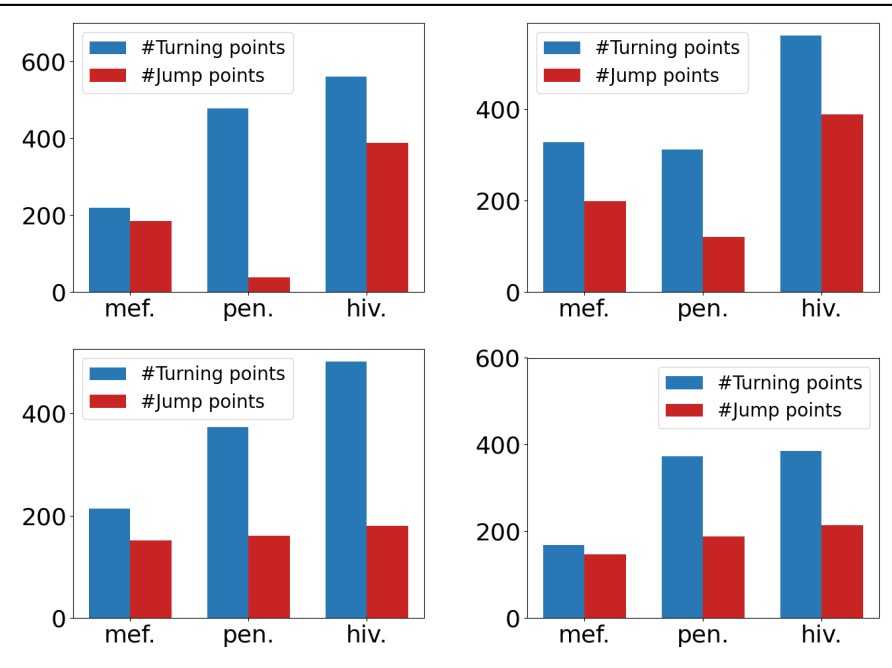

Figure 13: The number of different types of critical points in logistic regression with different parameters and regularizers. The first row of subfigures illustrate different critical points in the age path using linear SP-regularizer while the last row of subfigures show the number of different critical points in age-path using the mixture SP-regularizer.

| Dataset | Parameter $\alpha$ | Competing Methods | | | | | | Noise Level |
|---------|-----------|-------|-------|-------|-------|-------|-------|-------------|
| | | ACS | | MOSPL | | GAGA | | |
| music | 0.006 | 0.22 | 0.004 | 0.219 | 0.003 | **0.214** | **0.002** | 0.1 |
| music | 0.006 | 0.218 | 0.005 | 0.215 | 0.016 | **0.213** | **0.005** | 0.2 |
| music | 0.006 | 0.221 | 0.009 | 0.217 | 0.006 | **0.214** | **0.003** | 0.3 |
| music | 0.006 | 0.226 | 0.011 | 0.221 | 0.01 | **0.211** | **0.008** | 0.4 |
| music | 0.006 | 0.228 | 0.011 | 0.22 | 0.006 | **0.21** | **0.003** | 0.5 |
| music | 0.006 | 0.493 | 0.001 | 0.492 | 0.001 | **0.491** | **0.001** | 0.6 |
| music | 0.006 | 0.494 | 0.002 | 0.492 | 0.002 | **0.491** | **0.001** | 0.7 |
| music | 0.006 | 0.493 | 0.002 | 0.493 | 0.002 | **0.493** | **0.002** | 0.8 |

Table 14: Average generalization errors with the standard deviation in 20 runs under different noise levels. The top results in each row are in boldface.

| Dataset | Parameter C | Competing Methods | | | | | | Noise Level |
|---------|-----------|-------|-------|-------|-------|-------|-------|-------------|
| | | ACS | | MOSPL | | GAGA | | |
| mfeat-pixel | 1.000 | 0.946 | 0.018 | 0.967 | 0.007 | **0.980** | **0.006** | 0.1 |
| mfeat-pixel | 1.000 | 0.949 | 0.016 | 0.978 | 0.013 | **0.988** | **0.007** | 0.2 |
| mfeat-pixel | 1.000 | 0.937 | 0.054 | 0.962 | 0.281 | **0.980** | **0.015** | 0.3 |
| mfeat-pixel | 1.000 | 0.939 | 0.011 | 0.955 | 0.012 | **0.968** | **0.010** | 0.4 |
| mfeat-pixel | 1.000 | 0.941 | 0.008 | 0.962 | 0.005 | **0.972** | **0.008** | 0.5 |
| mfeat-pixel | 1.000 | 0.946 | 0.011 | 0.971 | 0.006 | **0.978** | **0.006** | 0.6 |
| mfeat-pixel | 1.000 | 0.934 | 0.013 | 0.955 | 0.005 | **0.959** | **0.005** | 0.7 |

Table 15: Average classification accuracy with the standard deviation in 20 runs under different noise levels. The top results in each row are in boldface.

| Dataset | Parameter | Competing Methods | | | | | | Noise Level |
|---------|-----------|-----------|-------|-------|-------|-------|-------|-------------|
| | C | ACS | | MOSPL | | GAGA | | |
| pendigts | 1.000 | 0.979 | 0.012 | 0.984 | 0.006 | **0.994** | **0.004** | 0.1 |
| pendigts | 1.000 | 0.980 | 0.003 | 0.991 | 0.008 | **0.998** | **0.004** | 0.2 |
| pendigts | 1.000 | 0.989 | 0.006 | 0.986 | 0.012 | **0.992** | **0.006** | 0.3 |
| pendigts | 1.000 | 0.980 | 0.009 | 0.990 | 0.005 | **0.996** | **0.006** | 0.4 |
| pendigts | 1.000 | 0.974 | 0.004 | 0.994 | 0.005 | **0.994** | **0.004** | 0.5 |
| pendigts | 1.000 | 0.984 | 0.007 | 0.986 | 0.010 | **0.996** | **0.004** | 0.6 |
| pendigts | 1.000 | 0.970 | 0.007 | 0.972 | 0.006 | **0.974** | **0.006** | 0.7 |

Table 16: Average classification accuracy with the standard deviation in 20 runs under different noise levels. The top results in each row are in boldface.

# E  Limitations and Broader Impact

**Limitations.**  The current framework GAGA is only applicable to the vanilla SPL paradigm, while some variants of SPL (*e.g.*, self-paced learning with diversity [2]) could have a group of hyperparameters more than the mere $\lambda$. In this instance, the original solution paths escalates into the solution surfaces, and made it harder to track the solutions. Another point is that our argument assumed that the target function is a biconvex problem, while many complex losses (*e.g.*, deep neural networks) could be the function of strongly non-convex.

**Broader Impact.**  The parties with limited computational resource may benefit from the efficiency of our proposed work. Meanwhile, the research groups that sensitive to age parameters in SPL may benefit from the robustness of our work. For social impact, our work improve the hyperparameter search and helps reducing the computation cost, which saves carbon emissions during the training process.

# F  Code Readme

This section explains how to use the code implementing the proposed GAGA. The codebase is available via github.com/diyang-lee/GAGA.

## F.1  Archive content

1. The fundamental implementation of our methodology GAGA is called `GAGA.py`, which includes all the functions used in our experiment. In more detail, they are named in the form of [GAGA]_[Model_Name]_[SPL_Regularizer], *e.g.*, GAGA_svm_linear.

2. The `Evaluation.py` provides functions for evaluating solution path from different models including ACS.

3. The `Input_Data.py` offers file-IO for all datasets used in our experiment.

4. The `ACS.py` provides ACS algorithm for solving SPL.

## F.2  Reproducing the results of the article

For the sake of quick and convenient reproducing experiments and checking our findings, we provide a demo named `main.py`. We use open-source datasets and provide file-IO functions along with detailed pre-processing pipeline for users. Any used datasets in our experiment can be easily downloaded them from the UCI and OpenML website and put them in the `datasets` folder. The required dependencies and running environment is recorded in `Environment.txt`.