# OpenReview forum: "GAGA: Deciphering Age-path of Generalized Self-paced Regularizer"
_NeurIPS.cc/2022/Conference — NeurIPS 2022 Accept_

### Official Review · Reviewer_sP8f · 2022-07-10

**Rating:** 6
**Confidence:** 2
**Soundness:** 2 fair
**Presentation:** 2 fair
**Contribution:** 3 good

**Summary:**

This paper proposes GAGA, an exact age-path algorithm to tackle general SP-regularizer for the self-paced learning (SPL) paradigm. The authors build the relationship between the GAGA framework and the existing theoretical analysis on SPL to provide a more in-depth understanding of the principle behind SPL.

**Questions:**

Major concerns about missing literature review and related works:
- Relation to curriculum learning (CL): according to [1], it seems the SPL is a sub-category of CL. However, there is no mention of the CL domain in this paper. The authors are recommended to conduct a comprehensive survey of the related works in CL in general, as well as other sub-categories of CL to discuss the relationships among them.
- Relation to the general robustness model under label noise: The paper claims the robustness of the proposed GAGA model, but did not conduct any comparative study to the SOTA robust model besides the SPL domain. For instance, robust SVM models such as [2, 3, 4], and robust lasso models such as [5] should also be considered as comparison methods in the experimental studies. In addition, a comprehensive literature review of the existing works about the general robustness models designed for handling noisy environments needs to be included as related works.

Major concerns about experimental validation:
- Concerns about the choices of the backbone model's parameters: how the backbone model parameters (e.g. alpha) are chosen in the experimental section for each dataset as shown in Table 2 and Table 3? Are they based on the baseline or GAGA? It will be unfair to compare the performance using the best parameter specifically selected based on GAGA on comparison methods.
- Concerns about the performance gain as compared to the existing works: the performance gain of the proposed GAGA is very marginal as compared to existing methods, as shown in Table 2 and Table 3. More specifically, less than 1% of improvement is observed in many datasets as compared with ACS and MOSPL, such as in pendigits, elevator, ailerons, and mfeat-pixel datasets, leading to no statistically significant conclusions of the results.
- Concerns about the choices of the noise ratios studied: In Table 2 and Table 3, only 20% and 30% noise ratios are selected and shown among different datasets without clear justification of the choices.  What is the rationale for choosing these specific numbers instead of applying a more comprehensive and well-spread range of the ratios (e.g. 20%, 40%, 60%, etc.)?

Reference:
[1] Soviany, P., Ionescu, R. T., Rota, P., & Sebe, N. (2022). Curriculum learning: A survey. International Journal of Computer Vision, 1-40.
[2] Biggio, Battista, Blaine Nelson, and Pavel Laskov. "Support vector machines under adversarial label noise." Asian conference on machine learning. PMLR, 2011.
[3] Yang, X., Tan, L., & He, L. (2014). A robust least squares support vector machine for regression and classification with noise. Neurocomputing, 140, 41-52.
[4] Chen, C., Yan, C., & Li, Y. (2015). A robust weighted least squares support vector regression based on least trimmed squares. Neurocomputing, 168, 941-946.
[5] Nasrabadi, N., Tran, T., & Nguyen, N. (2011). Robust lasso with missing and grossly corrupted observations. Advances in Neural Information Processing Systems, 24.


**Limitations:**

Not applicable. The authors did not provide any limitations or potential negative societal impact of their work.

**Strengths And Weaknesses:**

+ The proposed GAGA is the first exact age-path algorithm to tackle the general SP-regularizer for the self-paced learning (SPL) paradigm.
+ The authors build the relationship between the proposed GAGA framework and the existing theoretical analysis on SPL to provide a more in-depth understanding of the principle behind SPL.
+ The proposed GAGA is more computationally efficient than existing methods.
- Missing related works about the relation of the work to the general curriculum learning models
- Missing related works and experimental comparison to the general robustness model under label noise
- Missing justifications about many specific parameter choices in the experimental validation.

---

> ### Author Response · Authors · 2022-08-02
> **Response to Reviewer sP8f [2/2]**
>
> >Concerns about the performance gain as compared to the existing works.
>
> Thank you for your thorough review about experiments and your insightful feedback. Here we clarify the following points:
>
> 1. Our GAGA achieves the best practice of SPL implementation, while the competing baselines (conventional methods)  indeed track an approximation path of partial optimum in experiments. Assume the random seed is fixed, we have to point out that our GAGA can learn the entire solution spectrum and **is a deterministic algorithm**, however, numerical results of the competing baselines **is actually non-deterministic**. For instance, how to define the sequence of $\lambda$ in ACS search? It could be manual setting, Logarithmic growth, or equidistant series, etc. At present, **there's no unified specification, but they all compute the approximate solutions to our exact path**. Of course, we can choose another reasonable setting (but more beneficial to us) to enlarge the performance gap of baselines compared with results in the current manuscript, but we didn't do so.
>
> 2. The proposed GAGA not only focuses on the performance gain in simulation studies, **but also unifies the proposed analysis frameworks (about ours \& previous work) and provides a more in-depth understanding to the intrinsic mechanism** behind the SPL regime. (please see our Theorem 2 and Appendix B.1)
>
> 3. **Inherent defects of SPL/CL itself.** As summarized in a survey, curriculum learning may degrade data diversity and is not always bringing significant performance improvements. Please refer to Section 6 in the reference [1] that you provided. Since we are optimizing SPL itself rather than developing a novel learning paradigm, the experiments may also be affected by the shortcomings of SPL on some datasets.
>
> 4. When talking about the exact age-path algorithm, the performance gain, as compared with the earlier experiment, e.g., Table II in [f], **is also marginal** for around 1\% or 2\%. That's indeed consistent with what we have observed in fact.
>
> >The authors did not provide any limitations or potential negative societal impact of their work.
>
> **Now we have summarized limitations and societal impact of our framework in Appendix E.** (The revised parts are shown in red text.) We have updated our Checklist as well. Thank you for your reminder.
>
> **Reference**
>
> [a] Kumar, M., Benjamin Packer, and Daphne Koller. "Self-paced learning for latent variable models." NIPS 2010.
>
> [b] Meng, Deyu, Qian Zhao, and Lu Jiang. "A theoretical understanding of self-paced learning." Information Sciences. 2017.
>
> [c] Ghasedi, Kamran, et al. "Balanced self-paced learning for generative adversarial clustering network." CVPR. 2019.
>
> [d] Li, Hao, et al. "Multi-objective self-paced learning." AAAI 2016.
>
> [e] Gong, Maoguo, et al. "Decomposition-based evolutionary multiobjective optimization to self-paced learning." IEEE Transactions on Evolutionary Computation 2018.
>
> [f] Gu, Bin, et al. "Finding Age Path of Self-Paced Learning." ICDM 2021.
>
> [g] Sarker, Iqbal H. "Machine learning: Algorithms, real-world applications and research directions." SN Computer Science 2.3 (2021): 1-21.
>
> **We thank the reviewer sP8f again for detailed feedback. Please feel free to let us know if you still have any remaining concerns or questions, we will be happy to address them.**

---

> ### Author Response · Authors · 2022-08-02
> **Response to Reviewer sP8f [1/2]**
>
> Dear Reviewer sP8f, thank you for your review. **There potentially exists a misunderstanding about our motivations. We have also clarified your concerns or doubts in the evaluation part. We respectfully ask you to reconsider your rating.**
>
> >The paper claims the robustness of the proposed GAGA model, but did not conduct any comparative study to the SOTA robust model besides the SPL domain.
>
> Thank you for your constructive and valuable suggestions. **Our mission in this work is to achieve the best practice of the SPL itself by developing the state-of-the-art hyperparameter selection algorithm, rather than to chase the state-of-the-art robustness for specific ML models** like SVM or Lasso. As conceived in the original literature of vanilla SPL, an important motivation of SPL (as a variant of CL) is its robustness for confident sample selection [a]. As a matter of fact, SPL has achieved great success in lots of application tasks, the reason behind this is inseparable from its property of robustness [b, c]. As suggested by the reviewer, we conduct a series of comparative studies on the robust SVMs and robust Lasso models. Due to page limitation, we display this comparison part in the appendix, while we mention and cite the new baselines [2, 3, 4, 5] at line 320 in revision. **Our new results in Appendix D.6 demonstrate the robustness of GAGA against the state-of-the-art robust models**. Here we humbly ask the reviewer to have a look at the relevant subsections in the current manuscript. (The revised parts are shown in red text.) We also want to politely point out that this type of comparison doesn't exist in previous several (related) research papers [d, e, f].
>
> For the literature review about related works, we mentioned them (i.e., [2, 3, 4, 5]) in Section 5 at line 320 but we don't have more available spaces to talk about them in detail. Thank you for supplementing relevant work for us, and we will add these discussions in the final version.
>
> >The authors are recommended to conduct a comprehensive survey of the related works in CL in general, as well as other sub-categories of CL to discuss the relationships among them.
>
> Thank you for your supplement. We have updated the manuscript to describe the relationship between CL and SPL in Section 1 (see lines 21-26, revised parts are shown in red). Unfortunately, due to page limitations, the discussion of other sub-categories of CL (e.g., Teacher-student CL, Balanced curriculum (BCL)) could not be added to the current form of the manuscript, and we will add these discussions in the final version.
>
> >Concerns about the choices of the backbone model's parameters.
>
> The hyperparameters like $\alpha$ and $\lambda$ in the training objective are kept as same for both GAGA and competing methods for fairness, and **we didn't choose our hyperparameters deliberately or in some tricky ways.** (In fact, hyperparameters were randomly selected without inducing over-regularization and then fixed in experiments.) In order to make the experimental results more convincing, **we provide the results under more parameter settings in Appendix D.7**.  Here we humbly ask the reviewer to have a look at the relevant subsections in the current manuscript. (The revised parts are shown in red text.)
>
> In addition, the provided code also enables readers to try more hyperparameters. (e.g., parameter settings in specific research fields.)
>
> >Concerns about the choices of the noise ratios studied.
>
> Thank you for the feedback, the answer to why we selected the noise ratios as 20\% and 30\% is similar to the previous response.  Our empirical results verify the robustness of  GAGA under 20\%, 40\% and 80\% noise (please **refer to Figure 4** and line 298).
>
> What we would like to add is that the excessive noise ratio may be rare or unreasonable for **real-world** datasets [g]. Since SPL is more robust than the original learning paradigm and our GAGA can achieve the best practice of SPL (by selecting the best age parameter $\lambda$), our model will perform better when the noise ratio increases. **We also give the results under more noise ratio settings in Appendix D.7**.  Here we humbly ask the reviewer to read the relevant subsections in the current manuscript. (The revised parts are shown in red.)
>
> (We are running out of allowed characters, please see the second part [2/2])
>
> ---

---

> ### Author Response · Authors · 2022-08-08
> **Your thoughts on the response?**
>
> Dear Reviewer sP8f, inspired by your comments, we performed several experiments, including comparisons with SOTA robust models and a more comprehensive model's parameters. *What do you think of our response as well as new results?*

---

> > ### Comment · Reviewer_sP8f · 2022-08-09
> > **Thanks**
> >
> > The authors' response addressed most of the concerns with the additional discussions and experimental comparisons with the related works, thus I raise my score from 4 to 6.

---

### Official Review · Reviewer_ZV4b · 2022-07-11

**Rating:** 8
**Confidence:** 3
**Soundness:** 3 good
**Presentation:** 3 good
**Contribution:** 3 good

**Summary:**

This paper tackles the critical problem of how to optimally choose the age parameter and stop the increasingly learning process in self-paced learning (SPL). Grounded on the bi-convexity of the problem, the authors observe and prove that previous SPL methodologies are closely connected to the partial optimum of the SPL objective function. Thus, they reformulate the SPL paradigm as searching the partial optimum with optimal age parameter which is merely related to a single group of variables. Then, a generalized age-path algorithm that exactly traces the path of the partial optimum is proposed with the technique of ordinary differential equations (ODEs). Despite the general framework, two detailed algorithms on SVM and Lasso are discussed in the paper. Numerical results demonstrate the correctness and efficiency of the algorithm.

**Questions:**

1. It is expected to include [2] in the paper since it also provides insights to reveal the intrinsic mechanism behind SPL. Could the authors compare the differences between [2] and the proposed method in this paper?
2. In Section 4.1, the authors restrict the problem into the subspace generated by the basis $\{y_{i}\phi(x_{i})\}_{i=1}^{n}$, while the proof in Appendix B.4 adopts another way to convert the problem. It is a little confusing. Could the authors clarify more about these two statements? Is it possible to unify them?
3. What is the difference between Eq.(5) in the main body and Eq.(17) in the Appendix, so that the author placed them separately? In my understanding, they are both the detailed expansion of the KKT conditions of the original objective.
4. I noticed that $Gen$, $N_p$, and polynomial order $t$ are fixed for MOSPL. How did the authors choose these parameters in their experiments? Is it because they have been searched in advance, which is the best practice? If they are not fully studied, it is likely to damage the comparative fairness of the baseline.
5. Could the authors briefly summarize the limitations of this work?
***************
During the response period, the authors provided additional empirical results to support their experiments, and more discussions were supplemented in the Appendix. My concerns have been addressed. So I update my rating.

**Ethics Review Area:**

["I don’t know"]

**Limitations:**

Please refer to the above discussions and questions.
******************
During the authors' responses period, more discussions have been provided in the Appendix part.

**Strengths And Weaknesses:**

### Strengths:
1. The problem that this paper tries to address is significant in the SPL but lacks sufficient investigations both in the theoretical and practical aspects before. Unlike the existing studies, the framework derived in this paper is able to address most currently used self-paced regularizers (having explicit closed forms and following definitions in [1]) while owning theoretical guarantees.
2. The paper proposes a rigorous theoretical framework. All detailed proofs of all the results are given in the paper (except Theorem 5 in Appendix D). The experimental results are also fairly abundant in different views.
3. This paper provides a novel and interesting perspective to reconsider the previous SPL regime w.r.t the concept of partial optimum. The authors provide various discussions which are supported by both intuitions and theoretical results and dive much deeper above the concept itself.
4. The technique that utilizes the closed-form in alternative optimization to derive the age path is interesting. Since the problems of optimizing two groups of variables attract more and more focus (e.g., bi-level problems), I think this idea is of great potential to be studied further.


### Weakness:
1. It seems that the limitations of the proposed method are not explicitly discussed in the paper.
2. The theoretical framework and its derivation look quite complicated (to be rigorous and solid) and thus somewhat difficult to understand since there are many definitions, assumptions as well as reformulations. But the specific algorithms on the SVM and Lasso make up for this part in some sense.
3. The proposed framework is based on the earliest SPL regime, which involves only one hyperparameter. Hence the application scenarios of GAGA are restricted. The author should consider the proposed algorithm in more recent SPL formulations with more hyperparameters.

---

> ### Author Response · Authors · 2022-08-02
> **Response to Reviewer ZV4b**
>
> We thank the Reviewer ZV4b for very detailed feedback and for your appreciation of the paper. We have updated the manuscript to include the improvements suggested by the reviewer, which are detailed below together with our responses to your questions.
>
> >1. It is expected to include [2] in the paper since it also provides insights to reveal the intrinsic mechanism behind SPL. Could the authors compare the differences between [2] and the proposed method in this paper?
>
> We're sorry, seems that you missed references [1] and [2]. Thank you for supplementing relevant work for us, and we will add these references (after your updating) and discussions in the final version.
>
> >    2. Could the authors clarify more about the two statements in Section 4.1 and Appendix B.4?
>
> The former analysis in Section 4.1 is a theoretical view of the problem while the latter provides a practical approach. Thank you for the detailed feedback. **We have unified our statements in the current manuscript.** (The revised parts are shown in red text.)
>
> >    3. What is the difference between Eq.(5) in the main body and Eq.(17) in the Appendix?
>
> Eq.(5) is the detailed expansion  of KKT conditions that reveals the dynamics of $\hat{\boldsymbol{w}}$ and $\hat{\boldsymbol{t}}$, while Eq.(17) especially concentrates on  $\hat{\boldsymbol{t}}$ with fixed $\hat{\boldsymbol{w}}$ to indicate that $\hat{\boldsymbol{t}}$ indeed can be calculated given the corresponding $\hat{\boldsymbol{w}}$. The analysis on Eq.(17) provides an approach to compute $\hat{\boldsymbol{t}}$ through solving a linear system, which completes our framework.
>
> >    4. I noticed that $Gen$, $N_p$, and polynomial order $t$ are fixed for MOSPL. How did the authors choose these parameters in their experiments?
>
> First of all, we want to emphasize that the baseline method **MOSPL is insensitive to the initialization of parameters**. Please refer to: "Introduction" in [a] (Empirically)  or  "2.2 MOEA Techniques" in [b] (Theoretically).
>
> During the experiment we noticed that when some parameters of MOSPL were adjusted, such as $Gen$, $N_p$, it had little effect on the performance, but it significantly increase the computing time, which may lead the experimental results seem unfair (e.g., ~8 minutes for GAGA bur ~5 hours for MOSPL). **The table below shows some running results.** In experiments, we change the parameter $Gen$ and $N_p$, just like the majority of the Evolutionary and Intelligent Algorithms, MOSPL's performance improves quickly after $Gen = 200,N_p = 100$ but further improvement is small even multiple time is given (for example,$Gen = 1200,N_p = 400$, which basically means the computational cost is at least 23 times larger than that of $Gen = 200,N_p = 100$). Different values of $Gen$ and $N_p$ are chosen. The average generalization error and the standard deviation over multiple runs are listed in the last two columns.
>
> | Dataset            | $N_p$ | $Gen$  |  Mean  |  Std  |
> | :-------------- | :--: | :--: | :---: | :---: |
> | delta elevators |  5   |  5   | 0.669 | 0.013 |
> | delta elevators | 100  | 200  | 0.665 | 0.025 |
> | delta elevators | 150  | 300  | 0.666 | 0.009 |
> | delta elevators | 200  | 500  | 0.668 | 0.045 |
> | delta elevators | 250  | 800  | 0.668 | 0.043 |
> | delta elevators | 400  | 1200 | 0.673 | 0.051 |
> | ailerons        |  5   |  5   | 0.489 | 0.018 |
> | ailerons        | 100  | 200  | 0.489 | 0.023 |
> | ailerons        | 150  | 300  | 0.489 | 0.024 |
> | ailerons        | 200  | 500  | 0.489 | 0.015 |
> | ailerons        | 250  | 800  | 0.489 | 0.017 |
> | ailerons        | 400  | 1200 | 0.489 | 0.013 |
> | music           |  5   |  5   | 0.321 | 0.094 |
> | music           | 100  | 200  | 0.231 | 0.014 |
> | music           | 150  | 300  | 0.236 | 0.016 |
> | music           | 200  | 500  | 0.221 | 0.008 |
> | music           | 250  | 800  | 0.219 | 0.028 |
> | music           | 400  | 1200 | 0.223 | 0.016 |
>
> >    5. Could the authors briefly summarize the limitations of this work?
>
> Thank you for your valuable advice to improve our presentation. **We have summarized the limitations of our framework in Appendix E.** (The revised parts are shown in red text.) We have updated our Checklist as well.
>
> **Reference**
>
> [a]Li, Hao, et al. "Multi-objective self-paced learning." AAAI 2016.
>
> [b]Coello,  et al. "Evolutionary algorithms for solving multi-objective problems". Springer, 2007.
>
> **We thank the reviewer ZV4b again for the feedback. Please feel free to let us know if you still have any remaining concerns or questions, we will be happy to address them.**

---

### Official Review · Reviewer_Qt26 · 2022-07-16

**Rating:** 6
**Confidence:** 3
**Soundness:** 3 good
**Presentation:** 3 good
**Contribution:** 3 good

**Summary:**

This paper proposes an age-path algorithm (named GAGA) to tackle general SP-regularizer in self-paced learning.
During the optimization process, GAGA detects critical points via solving ODE.
Experimental results further verify the superiorty of GAGA over traditional SPL paradigms in multiple small-scale datasets.

**Questions:**

I hope the author could explain the motivation behind the proposed method more clearly.

Besides, using deep learning methods besides SVM and Lasso on some large-scale datasets to verify the effectiveness of the proposed method is more persuasive.

-----------
After reading the authors' response, I would increase the score from 5 to 6.

**Ethics Review Area:**

["I don’t know"]

**Strengths And Weaknesses:**

Strengths:

1. this studied problem is interesting and would attract attention from different machine learning topics


Weaknesses:

1. this paper is not easy to follow especially for the method section, many variables are not well defined ($m$ in Eq.(1), and $I_R$ is not mentioned in Section 3.1). Different sets in Line 160 are also explained with the right equation.

2. Overall, the motivation of the proposed method is not clear.

3. the experiment part is not enough to validate the effectiveness of the proposed method since most of these datasets in Table 1 are small and easy. Large-scale datasets like Dog-120 in [a] are suggested to be used in the experiment.

[a]. Wu, Xiaoping, et al. "BiSPL: Bidirectional self-paced learning for recognition from web data." IEEE Transactions on Image Processing 30 (2021): 6512-6527.

---

> ### Author Response · Authors · 2022-08-02
> **Response to Reviewer Qt26 [2/2]**
>
> **The table below show some running results on the Dog-120 dataset as you mentioned.** The classical SVM with the linear kernel is chosen as the base model and hyperparameter $C$ is set to 1.0 during the experiment. In implementation, we use the subsets of Dog-120 containing the most numerous 2, 10 or 40 kinds of dog pictures in original data, respectively. We flatten each picture to a 1-D array. The dimensionality  of each picture is unified as 600,000 by padding and truncation (due to the limited rebuttal  time and other considerations as mentioned in this comment above). We utilize the 'one vs all' strategy to form a binary (two-classes) classification task by selecting pictures of *Chihuahuas* class as positive samples, while the rest pictures are labeled as negative. **The top results in each row are shown in boldface.**
>
> | Dog Kinds | Original |  ACS  | MOSPL | GAGA  |
> | :-------: | :------: | :---: | :---: | :---: |
> |     2     |  0.831   | 0.855 | 0.887 | ***0.893*** |
> |    10     |  0.911   | 0.923 | 0.924 | ***0.931*** |
> |    40     |  0.945   | 0.961 | 0.966 | ***0.970*** |
>
> >using deep learning methods besides SVM and Lasso on some large-scale datasets to verify the effectiveness of the proposed method is more persuasive.
>
> Thank you for your valuable suggestions. After very careful consideration, we think that:
>
> 1. ***Our GAGA is a theory-guided framework with a strong mathematical guarantee***. In research papers, the deep learning models are typically regarded as a black box [a], and the theoretical analysis towards how neural network works is still under constant exploration by researchers [b]. Moreover, our ***GAGA is a deterministic ML algorithm*** while the optimization process of deep models has many randomness and uncertainty (e.g., the computation result of numerical solution is related to random seed initialization). ***It's precisely because we want to pursue theoretical guarantee that we didn't integrate the complex and non-trivial models (like deep learning) into our framework at the beginning stage.***
>
> 2. Our argument assumed that the target function is a ***biconvex problem*** (i.e., each subproblem regarding to model is convex), while the losses in deep neural networks could be the function of ***strongly non-convex*** [c]. That's unfortunately beyond the research scope of our current work.
>
> 3. We  noticed that the vanilla SPL brings the concept of biconvex target and uses mainstream ACS algorithm to handle the problem when it was proposed. The ACS algorithm is mature in the biconvex optimization theory, which has a relatively solid guarantee on the theory. In the mean time, ***the biconvex problem itself does have great application and research value***. For example, the SPL adopted by [d] on dictionary learning leads a biconvex objective. Hence ***we believe it's appropriate and reasonable to choose the biconvex setting, where our GAGA is cutting-edge and groundbreaking within the area.***
>
> **Reference**
>
> [a] Kingma, Diederik P., and Jimmy Ba. "Adam: A Method for Stochastic Optimization." ICLR 2015.
>
> [b] LeCun, Yann, Yoshua Bengio, and Geoffrey Hinton. "Deep learning." Nature 2015.
>
> [c] Damian, Alexandru, Jason Lee, and Mahdi Soltanolkotabi. "Neural Networks can Learn Representations with Gradient Descent." Conference on Learning Theory 2022.
>
> **We thank the reviewer Qt26 again for the feedback. Please feel free to let us know if you still have any remaining concerns or questions, we will be happy to address them.**

---

> ### Author Response · Authors · 2022-08-02
> **Response to Reviewer Qt26 [1/2]**
>
> Dear Reviewer Qt26, we thank you for the review and the positive assessment---below we address the specific concerns raised:
>
> >this paper is not easy to follow especially for the method section, many variables are not well defined ($m$ in Eq.(1), and $I_R$ is not mentioned in Section 3.1). Different sets in Line 160 are also explained with the right equation
>
> We attempt to derive the framework rigorously by using detailed symbols and formalized mathematical expressions, while it still seems not clear enough due to space constraints. Meanwhile, we try to make the paper self-consistent and easier to follow by considering specific problems where we always revisit the general framework and explicitly explain the meaning of symbols like $I_{\mathcal{R}}.$ **The $m$ in Eq.(1) refers to the number of regularization terms in the primal objective.** Although $I_{\mathcal{R}}$ is not directly used in Section 3.1, we define it here with all other definitions w.r.t. the index **to make the paper more ordered**. The partition sets defined in  Section 3 are in accordance with the convention in self-paced learning. In this paradigm, the weights of samples tend to be piecewise determined, which eventually leads to the partitioning of sample indexes according to their loss (as stated in line 160 and shown in Figure 3). The equation is illustrated as a reminder of this concept. We're very thankful for your feedback and will improve the presentations in our final version.
>
> >Overall, the motivation of the proposed method is not clear. I hope the author could explain the motivation behind the proposed method more clearly.
>
> The motivation part lies in line 40-50 and line 59-72 of the paper and we summarize them for you as follows:
>
> 1. The hyperparameter $\lambda$ in SPL is of vital importance but hard to select in implementations. The idea of solution path w.r.t. $\lambda$ (i.e., age-path) is a series of work to tackle the problem and has gained many achievements in this area.
>
> 2. **Current age-path algorithms are either limited to the simplest regularizer, or lack solid theoretical understanding as well as computational efficiency.** We propose the exact algorithm GAGA for *general* self-paced regularizers, which prevents a time-consuming hyperparameter tuning procedures.
>
> 3. Technically speaking, **age-path has the property of piecewise smoothness, in which the differential equations are very suitable to describe this type of structure**. In the meantime, the concept of partial optimum is well studied in the field of biconvex optimization, while we found that **SPL is a biconvex problem but  has never been connected to the partial optimum**. Additionally, ODE is a popular tool in the field of machine learning, also there are many literature studies on numerical ODE solving, prompting us to use this technology in GAGA.
>
> Unfortunately, we could not add more explanations about the motivation due to the page limitations. We're very thankful to your feedback and will improve the motivation part in our final version. **Please let us know if you have any doubts about the motivation of any part.**
>
> >the experiment part is not enough to validate the effectiveness of the proposed method since most of these datasets in Table1 are small and easy. Large-scale datasets like Dog-120 in [a] are suggested to be used in the experiment.
> [a]BiSPL: Bidirectional self-paced learning for recognition from web data. IEEE TIP2021.
>
> Thank you for supplementing relevant work for us and we've updated our reference at line 19. We didn't adopt the very large-scale datasets in original manuscript as:
>
> 1. **Our framework focuses on the theoretical frontier** (i.e., from the algorithmic perspective). Furthermore, in general machine learning practice, the model tends to be overfitted on small-scale datasets, which can better verify our algorithm in this scenario.
>
> 2. **The existing, mainstream SVM solvers do not have a well support of the parallelization**. Meanwhile, it's still difficult for existing machine learning packages (such as Scikit-learn) to adopt GPUs for acceleration. When the data size is large, it will significantly increase the solving time (of the subproblem in ACS) even if we own the rich computing resources.
>
> 3. We focus on the field of optimization. **The accelerations reflected in the limited data size can better reveal the effectiveness of our method.** (Generally speaking, on small-scale datasets, the difference between different competing algorithms may be very insignificant).
>
> (We are running out of allowed characters, please see the second part [2/2])
>
> ---

---

> ### Author Response · Authors · 2022-08-08
> **Does our response address your questions?**
>
> Dear reviewer Qt26, thanks again for your thoughtful review.
>
> Does our response address your questions? We would appreciate the opportunity to engage further if needed. We also kindly ask you to consider stronger support for the paper if your concerns have been addressed. Thanks!

---

> > ### Comment · Reviewer_Qt26 · 2022-08-08
> > **response to rebuttal**
> >
> > Thank you for the detailed response. Most of my concerns are well addressed, and I would increase the score from 5 to 6.

---

### Official Review · Reviewer_oJry · 2022-07-20

**Rating:** 6
**Confidence:** 3
**Soundness:** 3 good
**Presentation:** 3 good
**Contribution:** 2 fair

**Summary:**

This paper proposes a general age-path algorithm for various self-paced regularizers based on solving ordinal differential equations (ODE). Prior methods in self-paced learning (SPL) are prone to be underfitting and sensitive to noise and outliers, mostly due to the difficulty of choosing the optimal model age. Moreover, some existing solution path algorithms such as Lasso and SVM can not apply to SPL due to the optimization difficulty. To tackle these issues, this paper establishes a theoretical analysis based on the KKT condition and some fair assumptions and proposes a novel algorithm based on solving ODE. The methodology seems solid and the authors also introduce some guidance in practice.

**Questions:**

1. Please compare the proposed method with recent SOTA such as [1].
2. Please compare the training time of the proposed method with the baselines.
3. What is the meaning of 0 belonging to a scalar in Eqn.4?
4. Why are some results of the proposed method in bold in Table 2 and 3, even if they underperform the baselines?
[1]. Finding age path of self-paced learning. ICDM 2021.


After Rebuttal:
Thanks for the responses, which have addressed my concerns. Thus, I have raised my score from 4 to 6.

**Ethics Review Area:**

["I don’t know"]

**Limitations:**

Not applicable.

**Strengths And Weaknesses:**

Strengths:
This paper proposes an in-depth theoretical analysis of age-tracking in SPL. The proposed method incorporates KKT condition and ODE to advance the empirical bilevel solution of the prior method, leading to more stable, sample-efficient, and robust optimization.

Weaknesses:
The main concerns are clarity and experimental evaluation. In fact, I am confused by some notations and formulations. For example, 0 can not belong to another scalar in Eqn.4 (the right-hand side of Eqn.4 is a scalar). Secondly, the proposed method needs to solve an ODE in each iteration, as shown in Algorithm 2. It is important to discuss the training efficiency as solving an ODE in each iteration may be time-consuming. Moreover, the experimental results in Table 2 and 3 are quite misleading. The proposed method actually underperforms the baseline methods on many datasets but its results are in bold. Meanwhile, a comparison with the recent SOTA is also needed.

---

> ### Author Response · Authors · 2022-08-02
> **Response to Reviewer oJry [3/3]**
>
> Then the optimal $\hat{\boldsymbol{w}}=\sum\_{i=1}^{n}\hat{\alpha}\_{i}y\_{i}\phi(x\_{i})$, hence problem of SPL-SVM is transformed into solving equations merely related to $(\hat{\boldsymbol{ \alpha}},b).$ Moreover, the $\hat{\boldsymbol{\alpha}}\_{\mathcal{E}\_{Z}} - C \hat{\boldsymbol{t}}\_{\mathcal{E}\_{Z}} =\boldsymbol{0}\_{{\mathcal{E}\_{Z}}}$ reminds that the only undetermined component of $\hat{\boldsymbol{\alpha}}$ is the $\hat{\boldsymbol{\alpha}}\_{\mathcal{E}\_{Z}}$, and $b$ will be uniquely determined once $\hat{\boldsymbol{\alpha}}\_{\mathcal{E}\_{Z}}$ is chosen. As discussed in Section 3, all the index sets (e.g., $\mathcal{E}\_{Z}$) will retain the same within every smooth piece of age-path, which further indicates that $\hat{\boldsymbol{\alpha}}\_{\mathcal{E}\_{Z}}$ can be fixed on the whole piece of path while the KKT equations are maintained. In other words, a piecewise constant age-path algorithm is derived from here. Meanwhile, a critical point is met only when some index moves from $\mathcal{D}$ to $\mathcal{E}\_{P}$ since $v\_{i}$ equals to 0 for every $i\in \mathcal{E}\_{Z}\cup\mathcal{E}\_{N}$ excluding the necessity to concern about movement between these two sets.
>
> Consequently, the GAGA can be eventually simplified as a piecewise constant algorithm. The algorithm repeatedly computes the next index $i\in \mathcal{D}$ such that $g_{i}=0$ and changes the index sets accordingly to retain the KKT conditions, **which exactly consists with the algorithm derived in [1]**.
>
> **Reference**
>
> [a] Nagy, Dániel et al. "The art of solving a large number of non-stiff, low-dimensional ordinary differential equation systems on GPUs and CPUs." Communications in Nonlinear Science and Numerical Simulation 2022.
>
> [b] de Lima et al. "Accelerated solving of coupled, non-linear ODEs through LSTM-AI."  arXiv:2009.08278 (2020).
>
> [c] Dulmage A L et al. On the inversion of sparse matrices[J]. Mathematics of Computation, 1962, 16(80): 494-496.
>
> [d] Li S. Fast algorithms for sparse matrix inverse computations[M]. Stanford University, 2009.
>
> [e] Sun T et al. Sparse matrix inversion with scaled lasso[J]. The Journal of Machine Learning Research, 2013, 14(1): 3385-3418.
>
> [f] Sharma et al. (2013). A fast parallel Gauss Jordan algorithm for matrix inversion using CUDA. Computers \& Structures.
>
> **We thank the reviewer oJry again for the feedback. Please feel free to let us know if you still have any remaining concerns or questions, we will be happy to address them.**

---

> ### Author Response · Authors · 2022-08-02
> **Response to Reviewer oJry [2/3]**
>
> >solving an ODE in each iteration may be time-consuming.
>
> Thank you for feedback and we mentioned the complexity at line 238-240. Here we clarify the following points:
>
> 1. When at a new $\lambda$, assuming that the partition sets (e.g., $\mathcal{P}$) are known, even the fastest Newton’s method needs to solve the similar linear system multiple times until convergence. Detecting jumps of age-path by thresholding conditions takes an approximate  $\mathcal{O}(p)$ flops. The efficiency of GAGA lies in the fact that  ***no iterations are needed at the overwhelming majority values of*** $\lambda$ and it adaptively chooses step sizes to catch all events along the age-path.
>
> 2. Our equation (15) gives the explicit expression of gradient when solving ODEs, which substantiates the sparsity of matrix (i.e., contains some zero blocks) and leads the fundamental acceleration induced by Numpy / SciPy Library. While the major computational burden of ODE solving lies in the cost of matrix inversion, how to utilize the sparsity to accelerate the inversion calculation  ***has been extensively well-investigated*** in community [c, d, e]. For example, [d] derives an efficient and accurate FIND algorithm for certain sparse matrix computation of which the total cost is merely $\mathcal{O}(N^{\frac{3}{2}})$ for 2D problems. Meanwhile, parallel algorithms have light up the hope to make a breakthrough in this area with current high performance GPUs [f]. The [f] shows that the complexity of the traditional Gauss Jordan algorithm can decrease to $\mathcal{O}(n)$ on a massively parallel architecture, where all the $n^{2}$ computations can be done parallelly. Nowadays, more libraries capable of parallel computing are emerging, e.g. the pbdR packages in R.
> ***On the whole, we'd like to emphasize that the repeated ODEs solving process can efficiently speeds up integrating various factors. While the main focus of this work lies on the conduction and theoretical guarantee of the framework, we don't give too much clarification to this detail implementation issue due to the space limitation***.
>
> 3. Nowadays, the numerical ODEs solving library has been very efficient in computation, and the community is still in continuous exploration to pursuit the faster ODEs solving approaches [a, b].
>
> ***Proof of the equivalence***
>
> Specifically, [1] addresses the classical SVM and semi-supervised vector machine with the hard SP-regularizer. To be consistent with the paper we merely consider the classical SVM, while proof on the latter is similar.
>
> To begin with, the primal self-paced learning problem is formulated as:
> $$\min\_{\boldsymbol{w},b} \frac{1}{2}\\|\boldsymbol{w}\\|\_{\\mathcal{H}}^{2}+\sum\_{i=1}^{n}\left(C\max\left\lbrace  0,~1-y\_{i}(\langle \phi(x\_{i}),\boldsymbol{w}\rangle +b)\right\rbrace-\lambda v\_{i}\right),$$
>
> where $\lambda$ is the age parameter. Following the convention in the paper, we denote that $g_{i}=1-y_{i}(\langle \phi(x_{i}),\boldsymbol{w}\rangle +b)$,  $\ell_{i}=C\max\left\lbrace  0,~g_{i}\right\rbrace$, $\mathcal{E}=\\{1\leq i\leq n: \ell\_{i}<\lambda\\}$, $\mathcal{E}_{N}=\\{i\in\mathcal{E}:g\_{i}<0\\}, \mathcal{E}_{Z}=\\{i\in\mathcal{E}:g\_{i}=0\\}$, $ \mathcal{E}_{P}=\\{i\in\mathcal{E}:g\_{i}>0\\}$ and $\mathcal{D}=\\{1\leq i\leq n: \ell\_{i}\geq\lambda\\}.$ Using the same conduction in Section 4.1 and Appendix B.4, we can directly derive the simplified KKT equations.
>
> Formally, given a partial optimum $(\hat{\boldsymbol{w}},\boldsymbol{v}^{\*}(\hat{\boldsymbol{w}}))$ at $\lambda$, there exists $\hat{\boldsymbol{t}}=\left(\hat{t}\_{i}\right)^{n}\_{i=1}$ such that Eq.(20) holds. Denote $\hat{\boldsymbol{\alpha}}=C\boldsymbol{v}^{\*}\odot \hat{\boldsymbol{t}}$, then the last equations can be equivalently converted to equations w.r.t. $(\hat{\boldsymbol{\alpha}},b)$ as
> $$\begin{aligned}
>             \boldsymbol{y}^{T}\hat{\boldsymbol{\alpha}} &= \boldsymbol{0},\\\\
>             \boldsymbol{1}\_{\mathcal{E}\_{Z}}-Q\_{\mathcal{E}\_{Z}}\hat{\boldsymbol{\alpha}}-\boldsymbol{y}\_{\mathcal{E}\_{Z}}\hat{b}&=\boldsymbol{0},\\\\
>             \hat{\boldsymbol{\alpha}}\_{\mathcal{E}\_{P}} - C\boldsymbol{1}\_{\mathcal{E}\_{P}}&=
>             \boldsymbol{0}\_{{\mathcal{E}\_{P}}},\\\\
>             \hat{\boldsymbol{\alpha}}_{\mathcal{E}\_{Z}} - C
>             \hat{\boldsymbol{t}}\_{\mathcal{E}\_{Z}}&=\boldsymbol{0}\_{{\mathcal{E}\_{Z}}},\\\\
>             \hat{\boldsymbol{\alpha}}\_{\mathcal{E}\_{N} \cup \mathcal{D}}& = \boldsymbol{0}\_{\mathcal{E}\_{N} \cup \mathcal{D}},
> \end{aligned}$$
> where $Q=\boldsymbol{y}^{T}K\boldsymbol{y}$, $K=\left(k(x\_{i},x\_{j})\right)\_{1\leq i,j \leq n}$ is the kernel matrix and $k$ is the kernel function.
>
> (We are running out of allowed characters, please see the third part [3/3])
>
> ---

---

> ### Author Response · Authors · 2022-08-02
> **Response to Reviewer oJry [1/3]**
>
> Dear Reviewer oJry, thank you for your review. **There may exist misunderstandings about our claims and results. We respectfully ask you to reconsider your rating.** We address your questions/comments in order.
> >2. Please compare the training time of the proposed method with the baselines.
>
> The training time comparisons **are shown in Figure 5 and Figure 7**. We evaluated the running time between several algorithms under: (a.) different sample size, (b.) different interval length. More details can be found in Section 5. More results with different hyperparameters are shown in the Appendix D.7.
> >Moreover, the experimental results in Table 2 and 3 are quite misleading.
>
> Performances are measured by accuracy and generalization error for classification and regression, respectively. Please see the line 305-307. For Lasso regression task, we record the mean absolute error as the metric. The lower the number, the better the performance. (opposite to SVM). In our Tables 2 and 3, the first five rows are for SVMs and the last five rows are for Lasso. This can be identified by the different types of parameters (please see the second column). The results confirm GAGA's superiority and we believe that also answers your question 4. Sorry to make you misunderstood, we have added a dividing line in the middle of the tables to distinguish the two experimental models.
> >1. Please compare the proposed method with recent SOTA such as [1]. [1]. Finding age path of self-paced learning. ICDM 2021.
>
> Thank you for the paper. Only one specific SP-regularizer can be processed by [1], while GAGA is totally generalizable. Therefore, **the [1] is a special case of our framework.** (We have discussed [1] in paper line 61-63.) From a practical perspective, [1] and GAGA are indeed the same algorithm (when using hard SP-regularizer on SVMs). We first present the comparison results and then demonstrate the equivalence theoretically. Classical SVM with the linear kernel is used as the base model. The parameter C is selected from the best performance by using the grid search. The average accuracy and the standard deviation under different noise levels over multiple runs are shown below. The table suggests that the gained performance of [1] and GAGA are very close together.
>
> | Noise Level |  C    | APSPL |       |  GAGA |       | Dataset|
> | :---------: | ---- | ----: | :---: | ----: | :---: | :--: |
> |             |      | Acc |  Std  | Acc |  Std  |      |
> |     0.1     | 1.2  | 0.882 | 0.008 | 0.882 | 0.007 |  mfeat-pixel  |
> |     0.2     | 0.3  | 0.883 | 0.009 | 0.883 | 0.010 |  mfeat-pixel  |
> |     0.3     | 1.1  | 0.884 | 0.012 | 0.884 | 0.012 |  mfeat-pixel  |
> |     0.1     | 0.6  | 0.883 | 0.007 | 0.882 | 0.007 |  pendigits  |
> |     0.2     | 0.2  | 0.885 | 0.009 | 0.885 | 0.009 |  pendigits  |
> |     0.3     | 0.6  | 0.889 | 0.012 | 0.889 | 0.013 |  pendigits  |
> |     0.1     | 0.5  | 0.884 | 0.007 | 0.885 | 0.007 |  hiva  |
> |     0.2     | 0.5  | 0.884 | 0.008 | 0.884 | 0.008 |  hiva  |
> |     0.3     | 1    | 0.881 | 0.006 | 0.881 | 0.006 |  hiva  |
>
> The strict proof of the equivalence is given at the end of the comment.
>
> >3. What is the meaning of 0 belonging to a scalar in Eqn.4?
>
> Thank you for pointing this out. It's a typo. The $\partial$ in line 170 should be expressed as subdifferential (set of all subgradients), so the right hand of Eqn.4 is actually a set. This is the expression of KKT condition itself when there are no constrained conditions. We have corrected this typo in our revised version.
>
> (We are running out of allowed characters, please see the second part [2/3])
>
> ---

---

> ### Author Response · Authors · 2022-08-08
> **Thank you for upgrading your score.**
>
> Dear Reviewer oJry,
>
> Thanks for your timely reply and support for our work. Any further discussions are also welcome.

---

### Author Response · Authors · 2022-08-02
**Happy to Respond to Any Further Comments/Clarifications**

Dear Reviewers,

Thanks for the effort you put into reading and commenting on our work. You have given our work scores: 4, 5, 7 and 4, and have described our paper positively in the following way:

+ The problem that this paper tries to address is significant in the SPL but lacks sufficient investigations both in the theoretical and practical aspects before. It might also attract attention from different machine learning topics.

+ This paper provides a novel and interesting perspective and builds the relationship between the proposed GAGA framework and existing theoretical analysis. The methodology seems solid with detailed proofs of all the results and may bring a border impact on related fields (e.g. bi-level optimization).

+ The proposed GAGA is more computationally efficient than existing methods. Meanwhile, numerical results demonstrate the correctness and efficiency of the algorithm.

You have also raised some questions/concerns, to which we have replied in our detailed rebuttal. Specifically, we

+ Carried out more experiments (including: more noise ratios, more hyperparameters, more datasets) and in-detail analyses w.r.t the computational efficiency. Many concerns of the reviewers concentrate on the experimental validation and the efficiency of our method (e.g. the ODEs solving). ***Regarding these concerns, we first politely point out that there may exist some misunderstandings about our claims and results. From the theoretical perspective, we provide several related references and analyses to emphasize that the ODEs solving technique  can be indeed accelerated integrating various factors. Moreover, many acceleration schemes have been implemented in recent years. We gave less explicit clarification on this detailed implementation issue in the original manuscript since we mainly pay attention to the solid framework with strong theoretical guarantee. We attempted to stress the key points in the limited space. Empirically, we conducted plentiful supplementary experiments to dispel the key concerns raised by reviews.***

+ Improved presentation and typos (including the subdifferential in line 170)

+ Added more discussion on motivation and made explanation to the reviewers who may have misunderstood

+ Added some references of SPL applications, the CL and SOTA robust models.

+ Unified the notations of SVM in both main paper and proofs

+ Provided detailed proof of logistic regression

The supplementary proofs/experiments/references have also been carried out, ***please check them in our response part or in the new revision PDF***. We also kindly request you to consider stronger support for the paper if your concerns have been addressed. Please feel free to let us know if you still have any remaining concern, we will be happy to address them.

Thanks for your time and support!

Authors of Paper \#4954

---

### Meta-Review · Area_Chair_zG2R · 2022-08-25

**Recommendation:** Accept
**Confidence:** Certain

**Metareview:**

The paper received 4 positive reviews after the rebuttal. The technical concerns raised by the reviewers were addressed properly. Overall this work introduces a challenging and realistic setting that can be of large interest to the community working on self-paced learning.

**Award:**

No

---

### Decision · Program_Chairs · 2022-09-14

Accept